# Holimap: an accurate and efficient method for solving stochastic gene network dynamics

Chen Jia [1] & Ramon Grima [2] ✉

Gene-gene interactions are crucial to the control of sub-cellular processes but our understanding of their stochastic dynamics is hindered by the lack of simulation methods that can accurately and efficiently predict how the distributions of gene product numbers vary across parameter space. To overcome these difficulties, here we present Holimap (high-order linear-mapping approximation), an approach that approximates the protein or mRNA number distributions of a complex gene regulatory network by the distributions of a much simpler reaction system. We demonstrate Holimap's computational advantages over conventional methods by applying it to predict the stochastic time-dependent dynamics of various gene networks, including transcriptional networks ranging from simple autoregulatory loops to complex randomly connected networks, post-transcriptional networks, and post-translational networks. Holimap is ideally suited to study how the intricate network of gene-gene interactions results in precise coordination and control of gene expression.

Genetic regulation occurs through intricate interactions between a number of genes[1–4]. A gene "X" may express a protein which acts as a transcription factor (TF), promoting or inhibiting the RNA polymerase assembly on another target gene "Y" (or on itself) and thus regulating the extent that the latter is expressed[5]. These gene-gene interactions can be simply visualized as a directed graph with the genes being the nodes (vertices) and the directed edges (links) representing the interactions[6,7]. Networks inferred from gene expression data, commonly called gene regulatory networks[8], have been reconstructed by several methods[9–13]. The complex connectivity of these networks makes intuitive understanding of their dynamics challenging. Consequently, the construction, mathematical analysis, and simulation of models of gene regulatory networks are indispensable tools in a quantitative biologist's arsenal.

Several formalisms have been employed to predict gene regulatory network dynamics, including Boolean networks, ordinary differential equations (ODEs), and chemical master equations (CMEs)—for reviews covering these approaches and more, please see refs. 14,15. These approaches have various advantages and disadvantages. In Boolean networks, the expression of each gene is tracked by a binary variable and hence large networks can be examined in a computationally efficient way. A more refined description is provided by the use of ODEs, where the time-dependent concentrations of RNAs, proteins, and other molecules are predicted as a function of the rate constants of the reactions in the network[16,17]. An even more realistic description makes use of the CME approach where one predicts not only the mean expression levels of various genes but also the distributions of the discrete numbers of mRNAs and/or proteins measured across a population of cells[18]. This stochasticity has various sources (biological intrinsic and extrinsic noise, and technical noise introduced by experimental protocols), all of which lead to the large differences in gene expression observed from one cell to another[19–21].

Unfortunately, with an increasing level of sophistication and predictive power, simulations also rapidly become computationally expensive. Unraveling the stochastic dynamics of gene networks requires solving a set of coupled CMEs for the probability of the system being in each possible state. Since the number of states of a gene network is typically infinite, direct solution of these equations is

¹Applied and Computational Mathematics Division, Beijing Computational Science Research Center, Beijing, China. ²School of Biological Sciences, University of Edinburgh, Edinburgh, UK. ✉e-mail: Ramon.Grima@ed.ac.uk

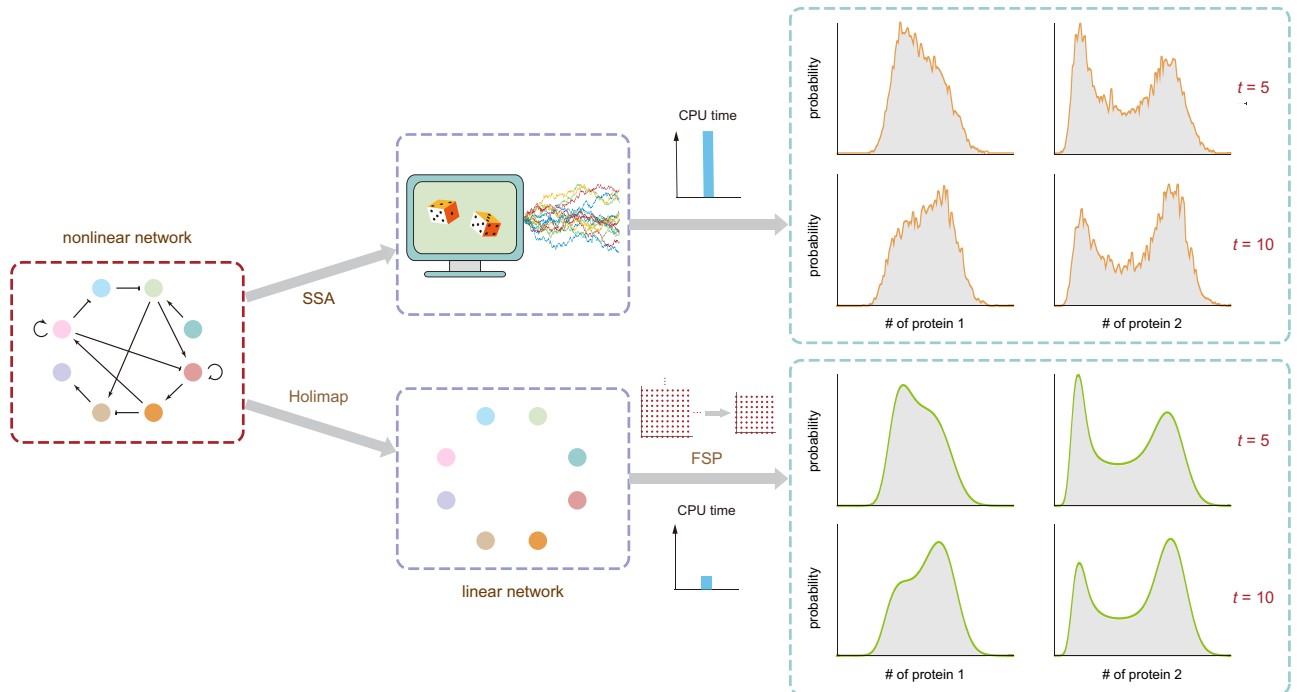

**Fig. 1 | Illustration of Holimap and its advantages over the SSA.** Holimap decouples gene-gene interactions in a nonlinear regulatory network and transforms it into a linear network with multiple effective parameters, some of which may be time-dependent. The time evolution of protein number distributions (for all genes) of the nonlinear network can be approximately predicted by solving the dynamics of the effective linear network using, e.g., FSP (the lattices in the lower row indicate that FSP truncates an infinite state space into a finite one and then solves the finite-dimensional CME). Compared with the conventional Monte Carlo approach (the SSA, whose two main stochastic steps are illustrated by dice), Holimap not only significantly reduces the CPU time, but it also yields an accurate, noise-free prediction of the protein number distributions.

impossible. The finite-state projection algorithm (FSP)[22] truncates the infinite state space to a finite one; this renders numerical solutions possible because we only need to solve a finite-dimensional CME. However, the immense number of states limits its applications to very small networks with one or two interacting genes. For larger networks with multiple interacting genes, Monte Carlo simulations based on the stochastic simulation algorithm (SSA)[23] become more practical. Specifically, given the current state of the system, the SSA generates two random numbers to predict the time when the next reaction event occurs and which particular reaction event will occur. The output is a number of statistically correct trajectories (molecule number versus time data), one for each cell, from which the copy number distributions of all biochemical species can be calculated. However, the issue remains that a large sampling size is typically required to obtain smooth distributions and hence the computational time can still be very considerable. For an introduction to simulation methods in stochastic biology, we refer the reader to refs. 24–26.

In this paper, we overcome the difficulties of conventional stochastic simulation methods for gene networks by devising an efficient approach—the high-order linear-mapping approximation (Holimap). The basic idea is to map the dynamics of a complex gene network with second or higher-order interactions (a system with nonlinear propensities and hence a nonlinear network) to the dynamics of a much simpler system where all reactions are first-order (a linear network). The reaction rates of this system are generally time-dependent and complex functions of the reaction rates of the original gene network and they are found by conditional moment-matching. The linear network has a much smaller state space than the nonlinear network which means that now simulation using FSP becomes feasible, leading to smooth distributions of protein numbers in a fraction of the time taken by SSA simulations. For an illustration of Holimap see Fig. 1.

The paper is structured as follows. The Holimap method is introduced by means of a simple autoregulatory feedback loop example where we show step-by-step how the approximation is constructed when second or higher-order interactions are only between a protein and a gene. The method is then extended to show the application to more complex networks with multiple protein-gene interactions and also to networks with gene product interactions such as those with RNA-RNA, RNA-protein, and protein-protein high-order reactions. By comparison with the SSA or FSP, we show that independent of the type of interactions in a gene network, Holimap provides highly accurate time-dependent distributions of protein or mRNA numbers over large swathes of parameter space including those regions where the system displays oscillatory or multistable dynamics. Finally, we show that the computation time of Holimap can be significantly reduced while maintaining its accuracy by devising a hybrid method which combines both Holimap and the SSA.

## Results

### Fundamental principles of Holimap illustrated by an auto-regulation example

Consider a simple autoregulatory feedback loop[27,28], whereby protein expressed from a gene regulates its own transcription (Fig. 2a). Feedback is mediated by cooperative binding of $h$ protein copies to the gene[29–32]. In agreement with experiments[33], protein synthesis is assumed to occur in bursts of random size $k$ sampled from a geometric distribution with parameter $p$, i.e., $\mathbb{P}(k=n)=p^n(1-p)$. Here $\sigma_b$ is the binding rate of protein to the gene; $\sigma_u$ is the unbinding rate; $\rho_b$ and $\rho_u$ are the burst frequencies of protein, i.e., the frequencies with which bursts are produced, when the gene is in the bound and unbound states, respectively; $d$ is the rate of protein degradation and dilution (due to cell division). The reaction system describes a positive feedback loop when $\rho_b > \rho_u$ (since in the case, binding of a protein increases

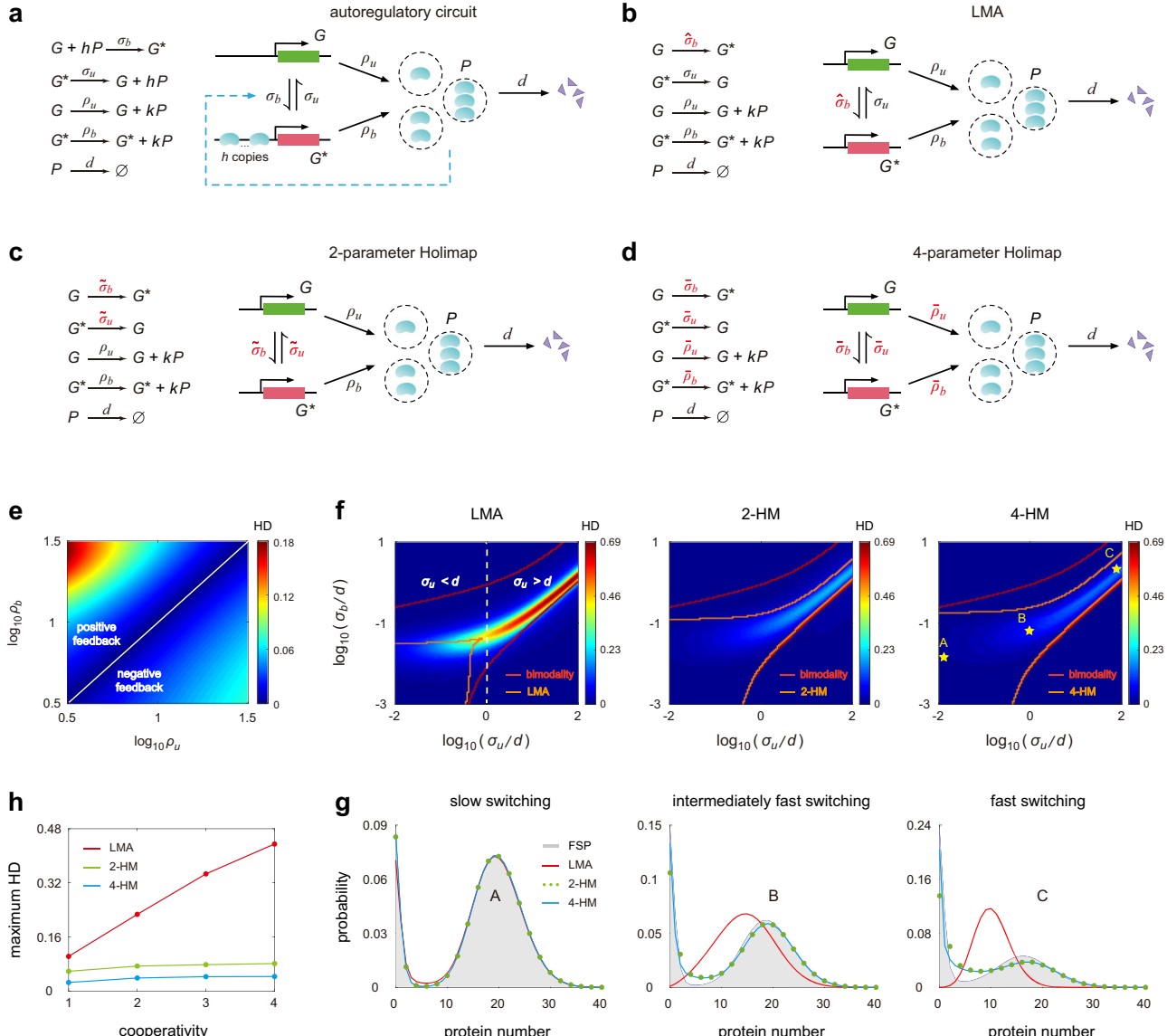

**Fig. 2 | Holimaps for autoregulatory gene networks in steady-state conditions.** **a** Stochastic model of an autoregulatory feedback loop, which includes bursty protein synthesis, protein decay, cooperative binding of protein to the gene, and unbinding of protein. **b** The LMA maps the nonlinear network to a linear one with effective parameter $\hat{\sigma}_b$. The high-order reactions $G + hP \rightleftharpoons G^*$ in the former are replaced by the first-order reactions $G \rightleftharpoons G^*$ in the latter. **c** The 2-HM maps the nonlinear network to a linear one with effective parameters $\tilde{\sigma}_u$ and $\tilde{\sigma}_b$. **d** The 4-HM maps the nonlinear network to a linear one with effective parameters $\bar{\sigma}_u, \bar{\sigma}_b, \bar{\rho}_u,$ and $\bar{\rho}_b$. **e** Heat plot of the HD for the LMA as a function of the protein burst frequencies $\rho_u$ and $\rho_b$. Here the HD for the LMA represents the Hellinger distance between the real steady-state protein distribution computed using FSP applied to the nonlinear system and the approximate protein distribution computed using the LMA. **f** Heat plots of the HDs for the LMA and Holimaps as functions of the unbinding rate $\sigma_u$ and

binding rate $\sigma_b$ (normalized by the decay rate $d$) when $\rho_b \gg \rho_u$. The red curves enclose the true bimodal region, i.e., the parameter region in which the protein number has a bimodal distribution, as predicted by FSP; the orange curves enclose the bimodal region predicted by the approximation method. The vertical white dashed line demarcates the region of $\sigma_u \geq d$, where the linear network given by the LMA can never exhibit bimodality, from the region of $\sigma_u < d$ where it can. **g** Comparison of the steady-state protein distributions computed using FSP, LMA, 2-HM, and 4-HM in different regimes of gene state switching. **h** The maximum HD as a function of the cooperativity $h$ for the LMA and Holimaps. Here the maximum HD is computed when $\sigma_u$ and $\sigma_b$ vary over large ranges, while other parameters remain fixed. See Supplementary Note 1 for the technical details of this figure. Source data are provided as a Source Data file.

its own expression) and describes a negative feedback loop when $\rho_b < \rho_u$ (binding of a protein decreases its own expression).

Let $p_{i,n}$ denote the probability of having $n$ protein copies in an individual cell when the gene is in state $i$ with $i = 0, 1$ corresponding to the unbound and bound states, respectively. To proceed, let $g_i = \sum_{n=0}^{\infty} p_{i,n}$ be the probability of observing the gene in state $i$ and let $\mu_{m,i} = \sum_{n=0}^{\infty} n(n-1)\cdots(n-m+1)p_{i,n}$ be the $m$th factorial moment of protein numbers when the gene is in this state. For simplicity, we first focus on the case of non-cooperative binding ($h = 1$). From the CME, it is straightforward to obtain the following time evolution equations for

the moments:

$$
\begin{aligned}
\dot{g}_0 &= \sigma_u g_1 - \sigma_b \mu_{1,0}, \\
\dot{\mu}_{1,0} &= \rho_u B g_0 - d\mu_{1,0} + \sigma_u(\mu_{1,1} + g_1) - \sigma_b(\mu_{2,0} + \mu_{1,0}), \\
\dot{\mu}_{1,1} &= \rho_b B g_1 - d\mu_{1,1} - \sigma_u \mu_{1,1} + \sigma_b \mu_{2,0}, \\
\dot{\mu}_{2,0} &= 2\rho_u B(\mu_{1,0} + Bg_0) - 2d\mu_{2,0} \\
&\quad + \sigma_u(\mu_{2,1} + 2\mu_{1,1}) - \sigma_b(\mu_{3,0} + 2\mu_{2,0}), \\
\dot{\mu}_{2,1} &= 2\rho_b B(\mu_{1,1} + Bg_1) - 2d\mu_{2,1} - \sigma_u \mu_{2,1} + \sigma_b \mu_{3,0},
\end{aligned}
\tag{1}
$$

where $g_1 = 1 - g_0$ and $B = \langle k \rangle = p/(1-p)$ is the mean protein burst size, i.e., the mean number of protein molecules produced in a single burst. For clarity, we have suppressed the explicit time dependence of all moments. Note that this system of equations is not closed, i.e., the equation for a moment of a certain order depends on moments of higher orders, and hence an exact solution is generally impossible. This difficulty stems from the nonlinear dependence on molecule numbers of the bimolecular propensity modeling protein-gene interactions[34].

In contrast, a linear gene network (one composed of only first-order reactions, i.e., the propensity of each reaction has a linear dependence on molecule numbers) is much easier to solve both analytically and numerically than a gene network with nonlinear propensities; for example, the moment equations are closed and thus can be solved exactly in this case. A basic idea of the linear-mapping approximation (LMA) developed in ref. 35 is to transform a complex nonlinear network into a linear one by replacing all second or higher-order reactions between proteins and genes by effective first-order reactions. Specifically, for the network in Fig. 2a, we replace the reactions $G + hP \rightleftharpoons G^*$ by $G \rightleftharpoons G^*$. The LMA maps the nonlinear network to the linear one shown in Fig. 2b, where the binding rate $\sigma_b$ for the former is replaced by the effective gene switching rate $\hat{\sigma}_b$ for the latter, while the other parameters remain unchanged. In the LMA, $\hat{\sigma}_b$ is chosen to be $\sigma_b$ multiplied by the conditional mean of protein numbers in the unbound gene state, i.e.,

$$\hat{\sigma}_b = \sigma_b \langle n | i = 0 \rangle = \frac{\sigma_b \mu_{1,0}}{g_0}, \tag{2}$$

where $g_0$ and $\mu_{1,0}$ can be calculated by a natural moment-closure method ("Methods")[35]. There are two approximations involved in the LMA: (i) in reality, the effective parameter $\hat{\sigma}_b$ should be stochastic rather than deterministic since it is proportional to the instantaneous protein number in the unbound state; (ii) any moment-closure method inevitably leads to some errors[36].

Next we propose an efficient method—Holimap, which we will show to perform much better than the LMA. There are two types of Holimaps. The first type is the 2-parameter Holimap (2-HM) which transforms the nonlinear gene network into the linear one illustrated in Fig. 2c, where both the binding and unbinding rates $\sigma_b$ and $\sigma_u$ for the former are replaced by the effective gene switching rates $\tilde{\sigma}_b$ and $\tilde{\sigma}_u$ for the latter. The remaining question is how to determine $\tilde{\sigma}_b$ and $\tilde{\sigma}_u$ so that the solution of the linear network accurately approximates that of the nonlinear one. For the linear network, the evolution of moments is governed by

$$\begin{aligned}
\dot{g}_0 &= \tilde{\sigma}_u g_1 - \tilde{\sigma}_b g_0, \\
\dot{\mu}_{1,0} &= \rho_u B g_0 - d\mu_{1,0} + \tilde{\sigma}_u \mu_{1,1} - \tilde{\sigma}_b \mu_{1,0}, \\
\dot{\mu}_{1,1} &= \rho_b B g_1 - d\mu_{1,1} - \tilde{\sigma}_u \mu_{1,1} + \tilde{\sigma}_b \mu_{1,0}, \\
\dot{\mu}_{2,0} &= 2\rho_u B(\mu_{1,0} + B g_0) - 2d\mu_{2,0} + \tilde{\sigma}_u \mu_{2,1} - \tilde{\sigma}_b \mu_{2,0}, \\
\dot{\mu}_{2,1} &= 2\rho_b B(\mu_{1,1} + B g_1) - 2d\mu_{2,1} - \tilde{\sigma}_u \mu_{2,1} + \tilde{\sigma}_b \mu_{2,0}.
\end{aligned} \tag{3}$$

The effective rates $\tilde{\sigma}_b$ and $\tilde{\sigma}_u$ are chosen so that the two systems have the same zeroth and first-order moment equations (for the latter, we mean the first-order moment when the gene is in the bound state). Matching the first and third identities in Eqs. (1) and (3), we find that $\tilde{\sigma}_b$ and $\tilde{\sigma}_u$ should satisfy

$$\begin{aligned}
\tilde{\sigma}_u g_1 - \tilde{\sigma}_b g_0 &= \sigma_u g_1 - \sigma_b \mu_{1,0}, \\
\tilde{\sigma}_u \mu_{1,1} - \tilde{\sigma}_b \mu_{1,0} &= \sigma_u \mu_{1,1} - \sigma_b \mu_{2,0}.
\end{aligned} \tag{4}$$

The remaining question is how to use these equations to obtain formulae for the effective rates. This can be done as follows: we first solve for $\tilde{\sigma}_b$ and $\tilde{\sigma}_u$ using Eq. (4) and then substitute these into Eq. (3) to obtain a set of closed moment equations. These equations can be

solved for the values of all zeroth, first, and second-order moments, i.e., $g_i$, $\mu_{1,i}$, and $\mu_{2,i}$. Finally substituting these into Eq. (4) gives the values of the effective parameters $\tilde{\sigma}_b$ and $\tilde{\sigma}_u$ for the linear network. See Supplementary Note 2 for a more detailed explanation of the Holimap algorithm.

In steady-state, the values of $\tilde{\sigma}_b$ and $\tilde{\sigma}_u$ are constants independent of time, and hence we can use the steady-state protein distribution of the linear network to approximate that of the nonlinear one—this can be computed analytically[37] or using FSP. When the system has not reached steady-state, the values of $\tilde{\sigma}_b$ and $\tilde{\sigma}_u$ depend on time $t$. In this case, we can use the time evolution of the linear network with time-dependent rates to predict that of the nonlinear one—while analytical solutions are not generally available in this case, the distributions can be efficiently computed using FSP.

In some regions of parameter space, the 2-HM may still not be accurate enough. To solve this problem, we devise a second type of Holimap—the 4-parameter Holimap (4-HM), which transforms the nonlinear network into the linear one illustrated in Fig. 2d. Here the binding rate $\sigma_b$, unbinding rate $\sigma_u$, and the protein burst frequencies $\rho_b$ and $\rho_u$ for the former are replaced by four effective parameters $\bar{\sigma}_b, \bar{\sigma}_u, \bar{\rho}_b$, and $\bar{\rho}_u$ for the latter, which can be determined by matching the moment equations for the two networks ("Methods"). Note that while for the 2-HM, we matched only the zeroth and first-order moments, for the 4-HM, we matched these and also the second-order moments. The 2-HM and 4-HM will be collectively referred to as Holimaps in what follows.

Thus far, we have only considered the case of $h = 1$. For the case of cooperative binding ($h \geq 2$), the Holimap approximation procedure can be similarly performed, except that higher-order moment equations need to be solved (Supplementary Note 2)—the algorithm for finding the effective parameters requires the solution of $(h+1)$-order moment equations. For example, when $h = 2$, third-order moment equations need to be solved and the effective parameters depend on the values of zeroth, first, second, and third-order moments. We emphasize that the computational cost of Holimap is mainly determined by the number of moment equations, $L$, to be solved. For autoregulatory loops, $L = 1 + 2h$ for the LMA and $L = 3 + 2h$ for Holimap. Note that the 2-HM and 4-HM have the same $L$.

The principles used to construct Holimaps for autoregulated networks can be used to obtain Holimaps for an arbitrarily complex network consisting of a system of interacting genes that regulate each other via positive or negative feedback. A flow chart of the Holimap algorithm for a general regulatory network can be found in Supplementary Fig. 1. The computational time of Holimap depends on the complexity of the network—an increased number of nodes (genes) or edges (regulatory reactions) results in an increased number of moment equations $L$ to be solved. In Supplementary Note 3, we prove that for a general network, $L$ scales polynomially with the cooperativity $h$ and scales exponentially with respect to the network size $M$ (number of genes).

## Applications to one-node (autoregulatory) networks

We now assess the performance of Holimap based on the Hellinger distance (HD) between the steady-state protein distribution obtained by applying FSP to the nonlinear network and the approximate distribution computed using the LMA and the two types of Holimaps. Note that while the direct application of FSP also leads to an approximate distribution, in effect, it can be considered exact since the error is very small provided the state space is truncated to a large enough value[22]. Here we choose the HD because it is bounded between 0 and 1; a visually accurate approximation is obtained when the HD $\ll 0.1$.

Figure 2e illustrates the HD for the LMA as a function of $\rho_u$ and $\rho_b$. Clearly, the LMA performs well when $\rho_u$ and $\rho_b$ are not very different from each other. However, it results in larger deviations from FSP when the protein burst frequency in one gene state is significantly larger

than that in the other. We also find that the LMA is much more accurate for negative feedback loops ($\rho_u > \rho_b$) than for positive feedback loops ($\rho_b > \rho_u$). In the LMA, the effective stochastic parameter $\hat{\sigma}_b$ is approximated by $\sigma_b$ multiplied by the conditional mean of protein numbers in the unbound state. Hence it must give rise to inaccurate approximations when protein noise in the unbound gene state is large. This is exactly what happens in the positive feedback case where the low synthesis rate in the unbound state results in a small conditional mean and thus large protein noise.

We next examine whether Holimap outperforms the LMA when it is applied to positive feedback loops. Figure 2f shows the HD against $\sigma_u/d$ and $\sigma_b/d$ for the LMA, 2-HM, and 4-HM when $\rho_b \gg \rho_u$. It is clear that the LMA (Fig. 2f, left) performs well when $\sigma_u$ and $\sigma_b$ are both small, but it becomes highly inaccurate when $\sigma_u$ and $\sigma_b$ are larger. The protein distribution can be unimodal or bimodal. The bimodal one is of particular interest because it indicates the separation of isogenic cells into two different phenotypes. In particular, we find that the LMA results in poor approximations when $\sigma_u \geq d$ and when the distribution is bimodal. This can be explained as follows. Recall that the LMA transforms a nonlinear network into a linear one with unchanged $\sigma_u$, which is commonly known as the telegraph model of stochastic gene expression[38]. In ref. [39], it has been proved that the telegraph model can produce a bimodal steady-state distribution only when both gene switching rates are smaller than the protein decay rate ($\sigma_u, \hat{\sigma}_b < d$). When $\sigma_u \geq d$, the linear network can never exhibit bimodality, while the bimodality in the nonlinear network can be apparent.

We emphasize that $\sigma_u \geq d$ is biologically relevant since in naturally occurring systems, protein is usually very stable[40] and hence its decay rate is often smaller than the rates of gene state switching. For example, in mouse fibroblasts, it has been measured[41] that the median protein half-life is 46 h and the mean cell cycle duration is 27.5 h; hence the mean decay rate of protein is estimated to be $d = (\log 2)/46 + (\log 2)/27.5\,\mathrm{h}^{-1} = 6.7 \times 10^{-4}\,\mathrm{min}^{-1}$. In the same cell type, the mean activation and inactivation rates for thousands of genes are estimated to be 0.002 min$^{-1}$ and 0.24 min$^{-1}$ [42]. In another study, the mean activation and inactivation rates are estimated to be 0.014 min$^{-1}$ and 0.17 min$^{-1}$[43]. Hence $\sigma_u \geq d$ is indeed satisfied for most genes.

In contrast to the LMA, both the 2-HM and 4-HM markedly reduce the HD values (Fig. 2f, center and right). The LMA has a maximum HD of 0.7, while for the two types of Holimaps, the maximum HDs are only 0.2 and 0.16. The 4-HM performs marginally better than the 2-HM in capturing steady-state protein distributions. We also compare the region of parameter space where bimodality is predicted to exist (region enclosed by the orange curves) with the actual region where bimodality manifests according to FSP (region enclosed by the red curves). We note that while the LMA fails to capture the bimodal region of the protein distribution, especially when $\sigma_u \geq d$, both the 2-HM and 4-HM capture the vast majority of the bimodal region. In summary, the deficiencies of the LMA for positive feedback loops are remedied by the use of Holimaps (Fig. 2g).

Finally, we examine how the cooperativity in protein binding affects the accuracy of various approximation methods. Figure 2h shows the maximum HD as a function of $h$ for the LMA, 2-HM, and 4-HM, where the maximum HD is computed when $\sigma_u$ and $\sigma_b$ vary over large ranges and other parameters remain fixed. Clearly, for the LMA, the maximum HD increases approximately linearly with respect to $h$ when $h \leq 4$; for Holimaps, the maximum HD is insensitive to $h$. Since TF cooperativity is the norm rather than the exception[5], our results suggest Holimap's accuracy remains high over the physiologically meaningful range of parameter values.

The results that we have presented assume steady-state conditions. However, the 2-HM can also accurately reproduce the time evolution of the protein distribution for nonlinear gene networks (Supplementary Fig. 2). The 4-HM is also accurate; however depending on parameter values, it may lead to numerical instability at short times,

which usually occurs when $\sigma_u$ and $\sigma_b$ are large for negative feedback loops (Supplementary Fig. 3). We did not observe numerical instability for the 2-HM. As a result, the 2-HM might be the preferable choice when dynamics is of major interest. In steady-state, while the improvement in accuracy of the 4-HM may be marginal, nevertheless since the two types of Holimaps require the solution of the same number of moment equations, the 4-HM is more advantageous when dynamics is not of interest.

## Applications to two-node networks with deterministic mono- and bistability

We next evaluate the performance of Holimaps when applied to study the steady-state behavior of two-node gene networks, where two genes regulate each other (Fig. 3a, left). Feedback is mediated by cooperative binding of $h_1$ copies of protein $P_1$ to gene $G_2$ and cooperative binding of $h_2$ copies of protein $P_2$ to gene $G_1$. Here $\sigma_{bi}$ and $\sigma_{ui}$ are the binding and unbinding rates for gene $G_i$, respectively; $\rho_{bi}$ and $\rho_{ui}$ are the synthesis rates of protein $P_i$ when the gene is in the bound and unbound states, respectively; $d_i$ is the degradation rate of protein $P_i$. For simplicity, we do not take protein bursting into account, although it can be included easily. Depending on whether $\rho_{ui} < \rho_{bi}$ or $\rho_{ui} > \rho_{bi}$ for $i = 1, 2$, there are four different types of effective system dynamics that constitute either a positive feedback or a negative feedback loop (Fig. 3b). For example, a toggle switch (two negative regulations)[44] corresponds to the case of $\rho_{u1} > \rho_{b1}$ and $\rho_{u2} > \rho_{b2}$. For two-node networks, Holimaps can be performed in a similar way as we have previously shown for autoregulatory loops, i.e., by replacing all protein-gene binding reactions by effective first-order reactions with new parameters and also allowing some of the other reactions to have different rate constants than those in the original network (Fig. 3a, center and right).

We first focus on a negative feedback loop without cooperative binding (Fig. 3c). Since the LMA performs well when the unbinding rate $\sigma_{ui}$ is much smaller than the degradation rate $d_i$, here we consider the case of $\sigma_{ui} \gg d_i$. We use the HD between the actual and approximate steady-state distributions of protein $P_1$ to test the accuracy of Holimap. Figure 3d illustrates the HDs for the LMA and 4-HM as functions of $\sigma_{b1}$ and $\sigma_{b2}$. We find that the network displays bimodality when $\sigma_{b1}$ is large and $\sigma_{b2}$ is small. This is surprising because in the literature there are two well-accepted origins for bimodality: (i) a positive feedback loop with ultra-sensitivity (type-I)[44] and (ii) slow switching between gene states (type-II), independent of the type of feedback loop[37]. Here the network is a negative feedback loop without cooperative binding, and thus there is neither a positive feedback loop nor ultra-sensitivity. Moreover, since both $\sigma_{u1}$ and $\sigma_{b1}$ are large, gene $G_1$ switches rapidly between the two states. Hence the bimodality observed is neither type-I nor type-II, and in the following, we refer to it as type-III bimodality.

From Fig. 3d, it is clear that the LMA performs poorly in this bimodal region. Again, the LMA cannot capture type-III bimodality since it transforms the nonlinear network into a linear one with unchanged $\sigma_{ui}$, which is unable to produce a bimodal distribution when $\sigma_{ui} \geq d_i$[39]. On the other hand, the 4-HM significantly reduces the HD values and performs exceptionally well in capturing the bimodal region (Fig. 3e). Here we do not show the 2-HM because it leads to similar results as the 4-HM except for a slightly larger HD value.

We next consider a toggle switch with cooperative binding, where two genes repress each other (Fig. 3f). Note that this is a positive feedback loop with ultra-sensitivity and hence it can produce deterministic bistability (type-I bimodality), which means that the corresponding system of deterministic rate equations (Supplementary Note 4) is capable of having two stable fixed points and one unstable point. Again, we only focus on the situation of $\sigma_{ui} \gg d_i$. Figure 3g illustrates the HDs for the LMA and 4-HM against $\sigma_{b1}$ and $\sigma_{b2}$. The yellow curve encloses the region of deterministic bistability, which is markedly smaller than the true bimodal region enclosed by the red curve. According to simulations, bimodality can be observed when

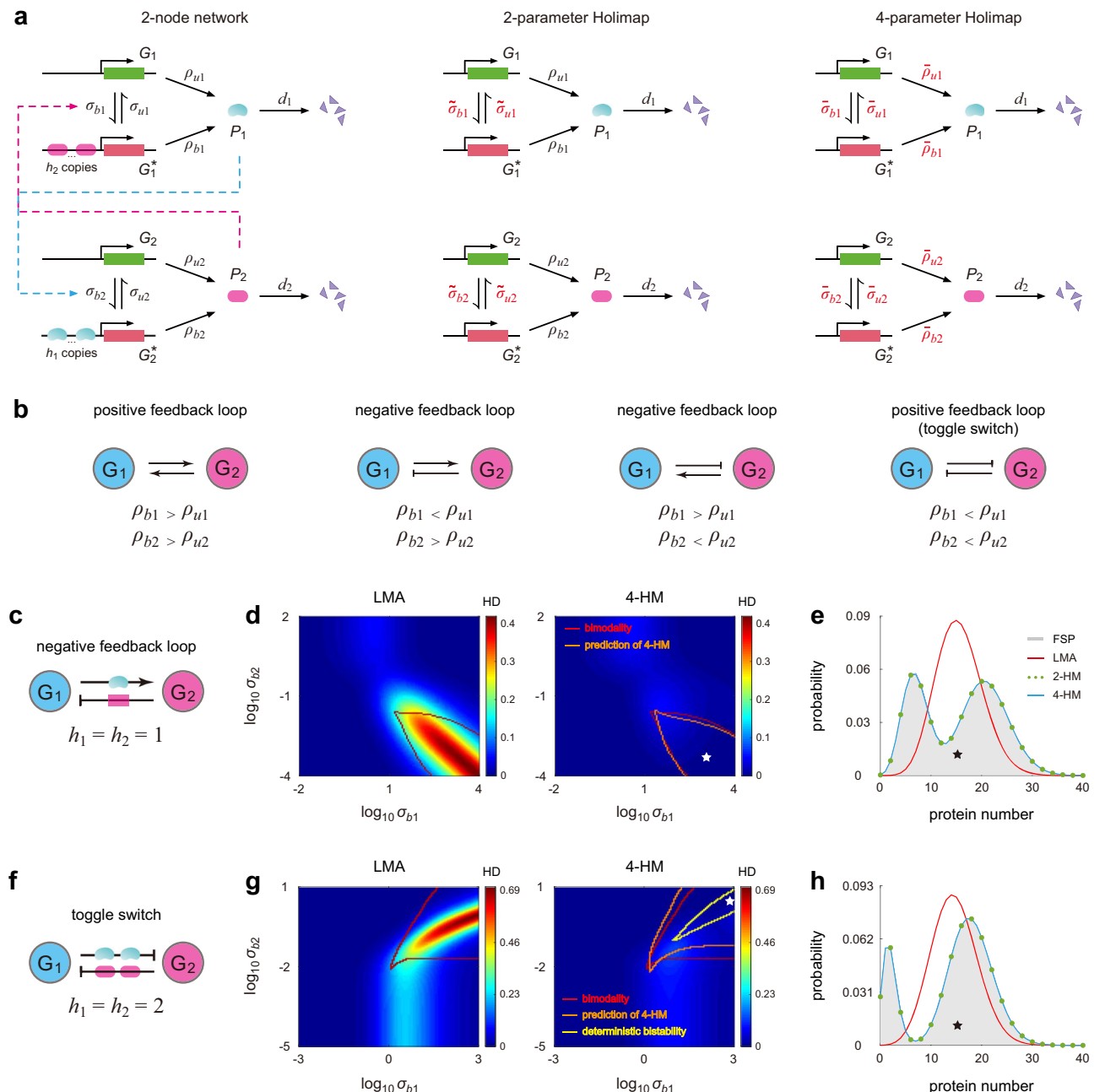

**Fig. 3 | Holimaps for two-node gene networks in steady-state conditions.**
**a** Illustration of the 2-HM and 4-HM for a two-node gene network, where two genes
$G_1$ and $G_2$ regulate each other. Feedback is mediated by cooperative binding of $h_1$
copies of protein $P_1$ to gene $G_2$ and cooperative binding of $h_2$ copies of protein $P_2$ to
gene $G_1$. **b** The two-node network can describe four different feedback loops,
according to whether $\rho_{ui} > \rho_{bi}$ or $\rho_{ui} < \rho_{bi}$ for $i = 1, 2$. **c** A negative feedback loop with
non-cooperative binding ($h_1 = h_2 = 1$). **d** Heat plots of the HDs for the LMA and 4-HM
as functions of the binding rates $\sigma_{b1}$ and $\sigma_{b2}$. Here the HD represents the Hellinger
distance between the real and approximate steady-state distribution of the number
of molecules of protein $P_1$. The red curve encloses the true bimodal parameter
region computed using FSP, and the orange curve encloses the bimodal region

predicted by Holimap. **e** Comparison of the steady-state distributions of protein $P_1$
computed using FSP, LMA, 2-HM, and 4-HM. **f** A toggle switch with cooperative
binding ($h_1 = h_2 = 2$). **g** Same as (**d**) but for the toggle switch. The yellow curve
encloses the parameter region of deterministic bistability, i.e., the region in which
the deterministic rate equations have two stable fixed points and one unstable fixed
point. **h** Same as (**e**) but for the toggle switch. Here the parameters are chosen so
that the system displays deterministic bistability. While we only focus on the dis-
tribution of protein $P_1$ in (**d**), (**e**), (**g**), and (**h**), the distribution of the second protein
$P_2$ is also accurately predicted by Holimap (Supplementary Fig. 3). See Supple-
mentary Note 1 for the technical details of this figure. Source data are provided as a
Source Data file.

both $\sigma_{b1}$ and $\sigma_{b2}$ are large. The LMA fails to reproduce the bimodal
distribution since $\sigma_{ui} \geq d_i$, as expected. The 4-HM not only successfully
captures the bimodal region (enclosed by the orange curve), but also
yields small HD values. The maximum HD for the LMA is as large as 0.7,
while it is only 0.13 for the 4-HM. In particular, in the deterministically
bistable region, both the 2-HM and 4-HM accurately predict the pro-
tein distribution while the LMA completely fails (Fig. 3h).

## Applications to three-node networks with deterministic oscillations
We now focus on three-node networks, where three genes regulate
each other in a cyclic manner (Fig. 4a, left). Feedback is mediated by
cooperative binding of $h_i$ copies of protein $P_i$ to gene $G_{i+1}$ for $i = 1, 2, 3$,
where $G_4 = G_1$. Again, depending on whether $\rho_{ui} < \rho_{bi}$ or $\rho_{ui} > \rho_{bi}$ for
$i = 1, 2, 3$, the network can be a repressilator (three negative

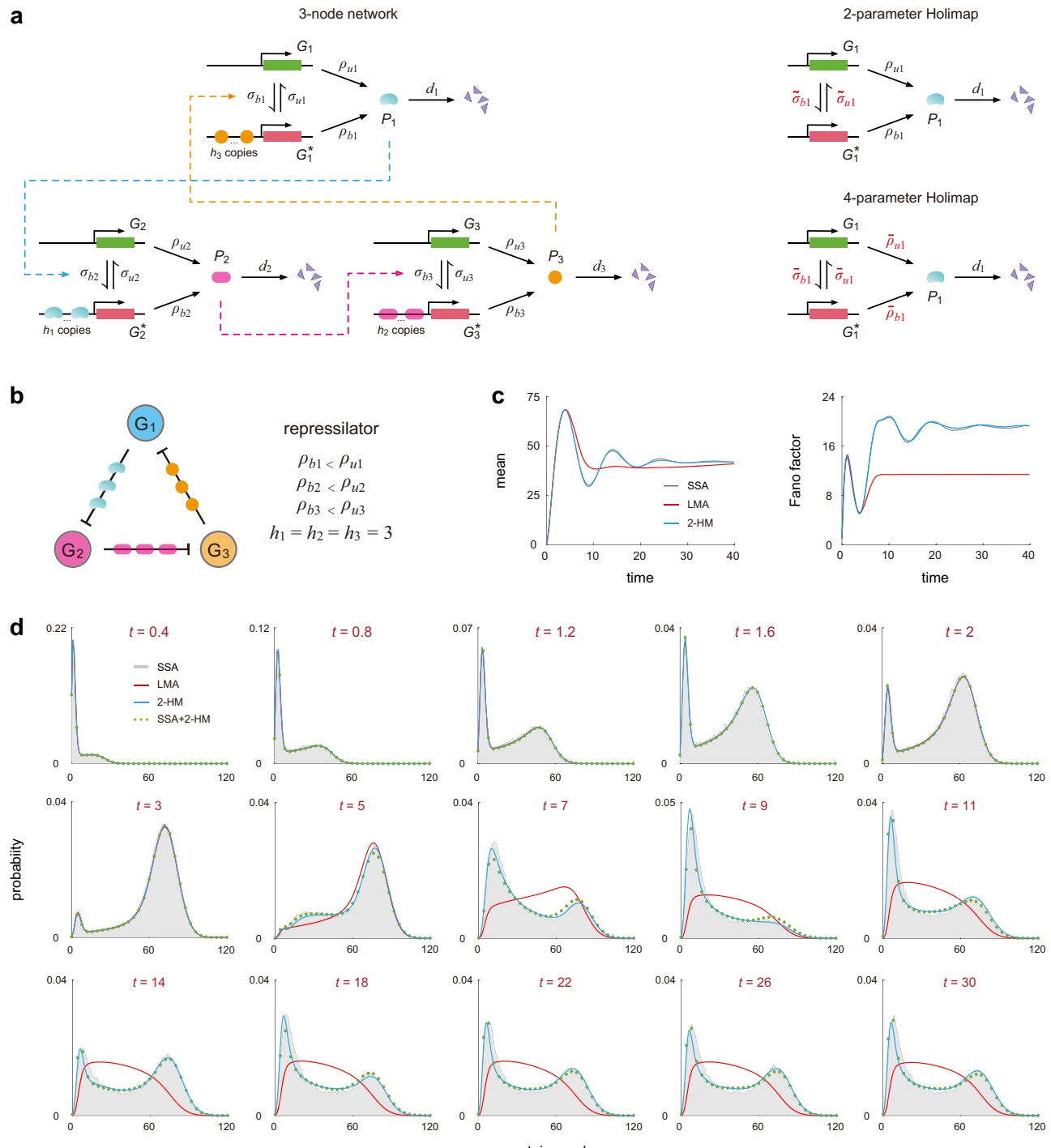

**Fig. 4 | Holimaps for three-node gene networks. a** Illustration of the 2-HM and 4-HM for a three-node gene network, where three genes regulate each other along the counterclockwise direction. Feedback is mediated by cooperative binding of $h_i$ copies of protein $P_i$ to gene $G_{i+1}$, where $G_4$ is understood to be $G_1$. **b** A repressilator with cooperative binding. Here the cooperativities are chosen as $h_i = 3$ for $i = 1, 2, 3$ such that the deterministic system of rate equations produces sustained oscillations. **c** Time evolution of the mean and Fano factor of fluctuations in the number of molecules of protein $P_1$ computed using the SSA (with $10^5$ trajectories), LMA, and 2-HM. **d** Comparison of the time-dependent distributions of protein $P_1$ at 15time points computed using the SSA (with $10^5$ trajectories), LMA, 2-HM, and hybrid SSA+2-HM (with 2000 trajectories). See Supplementary Note 1 for the technical details of this figure. Source data are provided as a Source Data file.

regulations)[45], a Goodwin model (one negative regulation and two positive regulations)[46], or a positive feedback loop[47].

As for previous examples, Holimap transforms the nonlinear network into a linear one (Fig. 4a, right). We now focus on the repressilator illustrated in Fig. 4b, where the cooperativities are chosen as $h_1 = h_2 = h_3 = 3$. Here high cooperativities are chosen since we require the corresponding deterministic system of rate equations

(Supplementary Note 4) to produce sustained oscillations. According to simulations, deterministic oscillations are not observed when $h_i \leq 2$. Figure 4c illustrates the oscillatory time evolution of the mean and Fano factor (the variance divided by the mean) of fluctuations in the number of protein $P_1$ computed using the SSA, LMA, and 2-HM. Note that here we do not consider the 4-HM because, as previously mentioned, it may cause numerical instability when computing time-

dependent distributions. The LMA fails to reproduce damped oscillations in the time evolution of the mean and Fano factor, while Holimap excellently captures these oscillations. Note also that the LMA significantly underestimates the variance of fluctuations and hence leads to a much smaller Fano factor in the limit of long times.

Figure 4d compares the time-dependent protein distributions computed using the SSA, LMA, and 2-HM. Interestingly, both the LMA and 2-HM accurately reproduce the protein distribution at small times ($t \leq 3$). However, the LMA fails to reproduce bimodality at intermediate and large times since it underestimates noise. In contrast, Holimap performs remarkably well in predicting the complete time evolution of the protein distribution.

Thus far, we have considered regulatory networks where each gene is regulated by one TF; however, many genes are regulated by a multitude of TFs which are often shared between multiple genes[48]. In Supplementary Note 5, we investigate gene networks with two TF binding sites. We show that Holimap performs excellently in capturing the protein distributions as well as the bimodal region, independent of the type of network topology and the type of TF binding (independent, positive cooperative, and negative cooperative binding[49]).

## A hybrid combination of SSA and Holimap provides highly efficient computation of complex gene network dynamics

The FSP and SSA are two widely used methods for solving the dynamics of stochastic chemical reaction systems. While FSP yields an accurate distribution, from a practical point of view, it is only applicable to small networks where protein numbers are not very large; for large networks, the size of the state space leads to an enormous computational cost[22]. The SSA can also be computationally expensive, particularly when the network has multiple reaction time scales[23]. When fluctuations are large, it can yield a non-smooth distribution, from which it is sometimes even difficult to determine the number of modes. To overcome this, a huge number of stochastic trajectories may be needed to obtain statistically accurate results. Holimap provides an accurate and smooth approximation of the protein distributions; however, it becomes computationally slow when the network is complex or the cooperativity is large since in these cases we have to solve a large number of moment equations. This raises an important question: is it possible to develop a highly efficient and accurate computation method of stochastic gene network dynamics that yields a smooth distribution?

To address this question, we propose a hybrid method that combines the SSA and Holimap. This method consists of three steps (Fig. 5a). First we use the SSA to generate a small number of trajectories (usually a few thousand trajectories are enough) from which we compute the steady-state or time-dependent sample moments of protein numbers. We then use the latter to compute the approximate effective parameters of the linear network. Finally, we use FSP to compute the protein distribution of the linear network with effective parameters to approximate that of the nonlinear one. For example, for the autoregulatory circuit illustrated in Fig. 2a, we substitute the sample moments obtained from the SSA into Eq. (4) to compute the approximate values of $\tilde{\sigma}_u$ and $\tilde{\sigma}_b$, and then use the marginal protein distribution of the linear network to construct the 2-HM of the nonlinear network. In other words, for Holimap, the determination of the effective parameters can be done independently of other computational methods while the hybrid method requires the running of the SSA.

This hybrid SSA + Holimap method is computationally much faster than the SSA because the number of trajectories needed to obtain good approximations to low-order moments is much less than that needed to obtain smooth protein distributions. It is also computationally less expensive than Holimap since there is no need to solve a large number of moment equations. To test this hybrid method, we compare the time-dependent distributions for the repressilator calculated using the SSA, LMA, 2-HM, and SSA + 2-HM (Fig. 4d). In Fig. 5b,

we also compare the CPU times and accuracy of these methods. The number of SSA trajectories $N$ needed for SSA + 2-HM is chosen such that the distributions obtained from $N$ trajectories and those obtained from $3N$ trajectories have an HD (averaged over all time points) less than 0.02, i.e., increasing the sample size will not substantially improve the approximation accuracy—a sample size of $N = 2000$ is sufficient to satisfy this criterion. Notably with almost the same CPU time, SSA + 2-HM yields distributions that are significantly more accurate than those obtained from only the SSA with the same number of trajectories—the HD for the former is only 0.04–0.06, while for the latter it is 0.11–0.13; here the distributions obtained from the SSA with $10^5$ trajectories are used as a proxy of ground truth when computing the HDs. We also note that SSA + 2-HM yields distributions that are practically as accurate as the 2-HM but are over 16 times faster (28 s vs 7 min 39 s).

To further test the accuracy of SSA + Holimap, we apply it to a random $M$-node gene network (Fig. 5c), where any pair of nodes has a probability of $2/M$ to be connected. This guarantees that each gene on average regulates two genes. When connected, each direct edge has an equal probability to be positive or negative regulation; autoregulation is also allowed. The details of the stochastic model are described in Methods. We then apply the 2-HM to transform the nonlinear random network into a linear one and then use 2000 SSA trajectories to estimate the effective parameters of the linear network. Figure 5d illustrates the CPU times and HDs against the number of nodes $M$ for SSA + 2-HM and the SSA with the same number of trajectories. Again an SSA with $10^5$ trajectories is used to generate a proxy of the ground truth distribution when computing the HDs. Clearly, the two methods yield almost the same CPU times that approximately linearly scale with $M$. This is because for SSA + 2-HM, almost all time is spent on simulating the SSA trajectories, while solving the linear network consumes very little time. However, compared with an SSA with 2000 trajectories, SSA + 2-HM gives rise to markedly lower HD values, which are insensitive to $M$.

## Generalization to networks with post-translational or post-transcriptional regulation

Thus far, we have showcased Holimap in transcriptional networks with protein-gene interactions. A crucial question is whether Holimap can also be applied to solve the dynamics of post-translational and post-transcriptional networks with complex protein-protein, protein-RNA, and RNA-RNA interactions. To see this, we first focus on two post-translational networks (Fig. 6a, b).

Figure 6a shows a two-node synthetic network with autoregulation and protein sequestration[50]. Here protein $P_i$ produced from gene $G_i$ regulates its own expression; the two proteins $P_1$ and $P_2$ can bind to each other and form an inactive complex $C$. We then devise a three-parameter Holimap (3-HM) which transforms the nonlinear gene network into the linear one shown in Fig. 6c. In principle, Holimap replaces all high-order interactions between genes, proteins, and RNAs by effective first-order reactions. We first replace the protein-gene binding reactions $G_i + h_i P_i \rightleftharpoons G_i^*$ by $G_i \rightleftharpoons G_i^*$ with effective parameters $\tilde{\sigma}_{ui}$ and $\tilde{\sigma}_{bi}$, and then we replace the protein-protein binding reaction $P_1 + P_2 \rightarrow C$ by $P_i \rightarrow \varnothing$ with effective parameter $\tilde{d}_i$. Again, using moment-matching, the three effective parameters $\tilde{\sigma}_{ui}, \tilde{\sigma}_{bi}$, and $\tilde{d}_i$ can be represented by low-order moments of the nonlinear network (Supplementary Note 7) and hence can be computed approximately using the SSA with a small number of trajectories. In this way, the hybrid SSA + Holimap can be applied to predict the dynamics of the nonlinear network.

Note that since Holimap replaces the binding reaction $P_1 + P_2 \rightarrow C$ by $P_1 \rightarrow \varnothing$ with a new parameter $\tilde{d}$, intuitively, one may deduce that this approximation is only valid when protein $P_2$ is very abundant compared to protein $P_1$ so that noise in protein $P_2$ number can be ignored. However, unexpectedly, we find that Holimap makes accurate predictions not only in this special case but also in scenarios where the

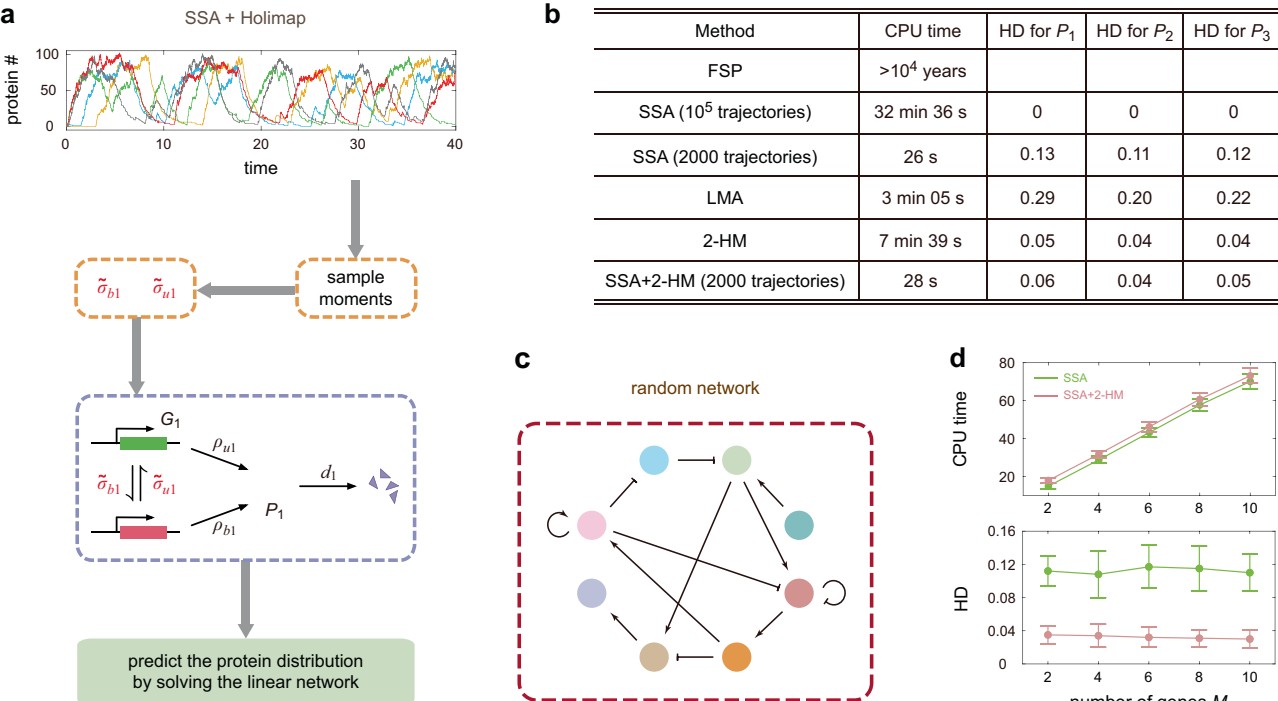

**Fig. 5 | A hybrid method combining the SSA and Holimap. a** SSA+Holimap serves as a highly efficient strategy to approximately solve the dynamics of complex stochastic gene networks. First, the SSA is used to generate a relatively small number of trajectories of the nonlinear network from which the steady-state or time-dependent sample moments are estimated. The low-order sample moments are then used to approximately compute the effective parameters of the linear network. Finally, the protein distributions follow directly by solving the dynamics of the linear network using FSP. **b** Comparison of the CPU times and the accuracy (measured by HDs at $t = 30$) for different methods. All methods were used to simulate the time-dependent protein distributions shown in Fig. 4d. The HD represents the Hellinger distance between the actual and approximate protein distributions. Here a proxy for the ground truth distribution is computed using the SSA with $10^5$ trajectories rather than FSP since the latter is computationally

infeasible (see Supplementary Note 6 for the estimation procedure for the CPU time required by FSP). **c** An illustration of a random $M$-node gene network, where each pair of nodes has a probability of $2/M$ to be connected. There are on average $2M$ directed edges for the network, each having an equal probability to be positive or negative regulation. **d** Comparison of the CPU times and the accuracy (measured by HDs averaged over ten-time points and overall proteins) against the number of nodes $M$ for SSA+2HM and the SSA with the same number of trajectories. Here the number of trajectories needed for both SSA+2HM and SSA is chosen as $N = 2000$. Both methods were used to simulate the time-dependent distributions for a random network. Data are presented as mean values +/− standard deviations for five different random networks. See "Methods" for the technical details. In (**b**) and (**d**), all simulations were performed on an Intel Core i9-9900K processor (3.60 GHz).

two proteins interact at comparable concentrations and where $P_2$ is very scarce compared to $P_1$ (Supplementary Fig. 5). This again confirms the high accuracy of Holimap over large regions of parameter space.

As another example of post-translational regulation, we consider a gene network with autoregulation and protein phosphorylation (Fig. 6b), which has been used to account for circadian oscillations in *Drosophila* and *Neurospora*[51]. Here the free protein $P$ can be reversibly phosphorylated into the forms $P_1$ and $P_2$, successively. The latter form $P_2$ can bind to the gene and regulate its expression. Both phosphorylation and dephosphorylation are enzyme-catalyzed and are described using Michaelis-Menten kinetics. Holimap can also be applied to this network, where protein-gene interactions are replaced by the switching reactions $G \rightleftharpoons G^*$ with effective parameters $\tilde{\sigma}_u$ and $\tilde{\sigma}_b$, and the complex post-translational regulation is replaced by the degradation reaction $P \rightarrow \varnothing$ with effective parameter $\tilde{d}$ (Fig. 6c and Supplementary Note 7).

Furthermore, we apply Holimap to two post-transcriptional networks (Fig. 6d, e). Figure 6d illustrates a gene network with auto-regulation and mRNA degradation control[52]. Here the enzyme can convert between an inactive form $E$ and an active form $E^*$. The degradation of the mRNA of interest can occur spontaneously with rate $d$ and can be catalyzed by the active form of the enzyme with rate $\alpha$. Holimap transforms the nonlinear network into the linear one shown in Fig. 6f by removing all high-order interactions between molecules. In particular, the enzyme-catalyzed degradation reaction $M + E^* \rightarrow E^*$ is

replaced by the effective degradation reaction $M \rightarrow \varnothing$ with new parameter $\tilde{d}$ (Supplementary Note 7).

Figure 6e illustrates another network with microRNA-mRNA interactions, which has been shown to be capable of producing complex emergent behaviors such as bistability and sustained oscillations[53]. Here the mRNA of interest, expressed from gene $G_1$, has two microRNA binding sites. The microRNA, produced from gene $G_2$, can bind to its mRNA target and form two inactive complexes $C_1$ (only one binding site is occupied) and $C_2$ (both binding sites are occupied). The free mRNA and microRNA are degraded with rates $d_1$ and $d_2$, respectively. Once the complex $C_1$ ($C_2$) is formed, the mRNA and microRNA are degraded with rates $a_1$ ($b_1$) and $a_2$ ($b_2$), respectively. The mRNA dynamics for this network can also be predicted by Holimap, which replaces the complex post-transcriptional regulation by the effective reaction $M \rightarrow \varnothing$ with new parameter $\tilde{d}$ (Fig. 6f and Supplementary Note 7).

Note that for transcriptional networks, Holimap does not change the degradation rate; however, for post-transcriptional and post-translational networks, both the binding/unbinding rate and degradation rate need to be modified. To test the accuracy of the three-parameter Holimap, we compare the time-dependent distributions for the above four gene networks computed using the SSA with $10^5$ trajectories, SSA with 2000 trajectories, and hybrid SSA + Holimap with 2000 trajectories (Fig. 6g). Clearly, SSA+Holimap is accurate for all networks. In particular, the distributions predicted by SSA + Holimap with a small number of trajectories have almost the same accuracy as

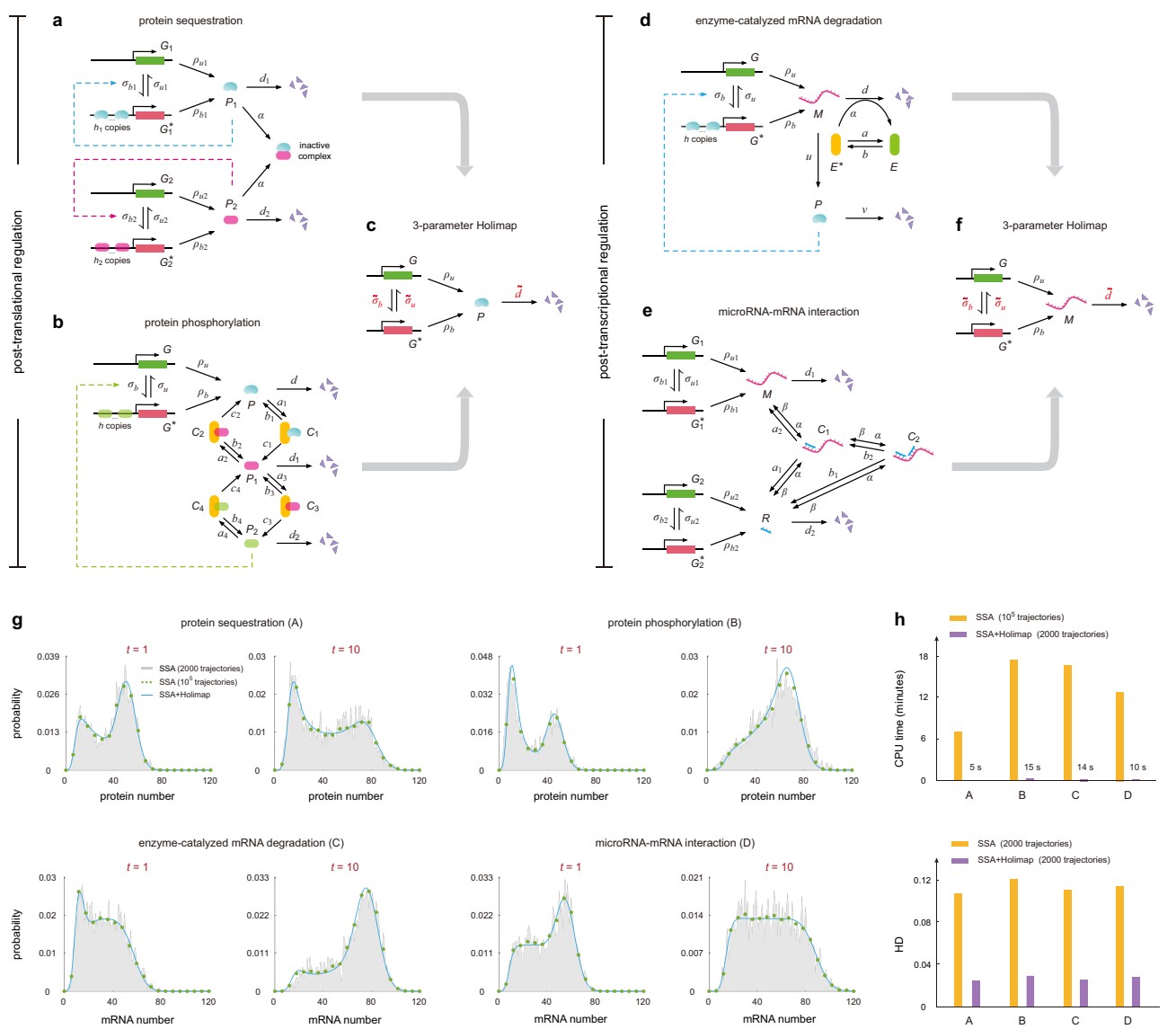

**Fig. 6 | Holimap for post-translational and post-transcriptional networks.** **a**, **b** Post-translational networks. **a** Network with autoregulation and protein sequestration[50]. **b** Network with autoregulation and protein phosphorylation[51]. **c** Three-parameter Holimap for post-translational networks. **d**, **e** Post-transcriptional networks. **d** Network with autoregulation and mRNA degradation control[52]. **e** Network with microRNA-mRNA interactions[53]. Here $\alpha$ is the binding rate of microRNA to its mRNA target and $\beta$ is the unbinding rate. **f** Three-parameter Holimap for post-transcriptional networks. All high-order interactions between genes, proteins, and RNAs are replaced by effective first-order switching and degradation reactions. **g** Comparison of the protein distributions for the four networks at two time points computed using the SSA with $10^5$ trajectories, SSA with 2000 trajectories, and SSA + Holimap (with 2000 trajectories). **h** Comparison of the CPU times and the accuracy (measured by HDs averaged over ten-time points) of the SSA and SSA + Holimap for the four networks. The distributions obtained from the SSA with $10^5$ trajectories are used as a proxy of ground truth when computing the HDs. See Supplementary Note 1 for the technical details of this figure. Source data are provided as a Source Data file.

those predicted by the SSA with a huge number of trajectories (HD < 0.03) while the CPU time is reduced by over 60 fold (Fig. 6h).

## Discussion

In this paper, we have constructed a computational method, Holimap, for the accurate and efficient prediction of the protein/mRNA number distributions of a general gene regulatory network. We have showcased the method by applying it to a variety of networks including transcriptional networks with protein-gene interactions, post-translational networks with protein-protein interactions, and post-transcriptional networks with protein-RNA or RNA-RNA interactions. For transcriptional networks, we have tested Holimap in simple autoregulatory loops where a gene influences its own expression, two-gene systems such as the toggle switch, three-gene systems such as the

repressilator, and complex randomly connected networks with numerous interacting genes. Notably, we have demonstrated that a hybrid method that uses both Holimap and the SSA leads to much more accurate distributions than solely using the SSA, with practically no increase in the CPU time and high accuracy that is independent of the number of interacting genes in the network.

We devised three types of Holimaps—the 2-HM, 3-HM, and 4-HM—all of them decoupling gene-gene interactions in a nonlinear regulatory network and transforming it into a linear one with multiple effective parameters. The 2-HM and 4-HM apply to transcriptional networks, while the 3-HM is only applicable to post-translational and post-transcriptional networks. The 4-HM is more accurate than the 2-HM, although the improvement in accuracy is marginal. Depending on parameters, the 4-HM may lead to numerical instability at short times.

Hence the 4-HM is preferred if our aim is to compute the steady-state distribution, and the 2-HM is a preferable choice if our aim is to compute the time-dependent distribution. The two types of Holimaps require the solution of the same number of moment equations and hence give rise to similar CPU times. Since the number of equations to be solved increases exponentially with the network size, the standard Holimap is only recommended when the scale of the network is not too large. For medium and large-scale networks, the hybrid SSA+Holimap approach is more advantageous since it significantly reduces the CPU time while maintaining high accuracy.

Some of the advantages of our method over other common approximations in the literature are as follows: (i) Holimap does not sacrifice the discrete nature of molecular reactions since the approximate distributions are solutions of the CME of the effective linear network. This is unlike many common methods that achieve a speed increase by making use of a continuum approximation of the CME such as the Fokker-Planck / Langevin equations[54,55] or partial integrodifferential equations[56,57]; (ii) Holimap does not assume the protein number distribution to be of a simple type such as the Gaussian, Poisson, Lognormal or Gamma distributions, as commonly assumed by conventional moment-closure methods[58,59]—the solution of the linear network that Holimap utilizes is very flexible and spans a very large number of possible distribution shapes including those with multiple modes and significant skewness. For example, if each gene in a complex regulatory network switches between a number of states for which only one is active, then Holimap approximates the protein distribution for each gene by that of a multi-state gene expression model with no regulatory interactions (Supplementary Note 5) for which the analytical steady-state solution is known to be a generalized hypergeometric function[60,61], which includes a large number of special functions as special cases.

Our hybrid SSA+Holimap method shares some similarities with neural network-based approaches[62], which can also be used to predict complex gene network dynamics. The former uses the SSA to generate the sample moments which are then used to compute the values of effective parameters, while the latter uses the SSA to train the surrogate neural network model. While both methods can accurately capture the protein/mRNA distribution, our method outperforms the neural network-based ones in three aspects: (i) while neural network models perform well in the parameter ranges which are used to train the surrogate model, their extrapolation ability is usually weak. Our method is mechanism-based and provides accurate results over wide parameter ranges; (ii) neural network-based methods require a very long time to train the surrogate model. When the network is complex, the training time may take tens of hours to several days and may also require multiple rounds of hyperparameter tuning. In contrast, Holimap avoids the long training time; (iii) neural network models have good predictive ability but their learned approximation does not typically have a clear biophysical interpretation. Holimap transforms the complex nonlinear network into a linear one which not only has a clear physical meaning but also allows an approximative analytical solution. In addition, SSA + Holimap can be combined with neural network-based methods to increase the speed and accuracy of the latter. Since SSA + Holimap can be used to generate distributions comparable in accuracy to those from the SSA with a much larger number of trajectories, it follows that SSA + Holimap can reduce the time to generate an accurate training dataset as input to the neural network.

The main limitation of the present method is that there are no analytical guarantees that the effective parameters of the linear network are positively definite for all times. Nevertheless, for all examples using the 2-HM and 3-HM in this paper, we have numerically found this to be the case and hence we are confident that the linear network obtained by the 2-HM or 3-HM procedure is generally physically interpretable. In contrast, we observed that the 4-HM procedure can occasionally give rise to negative parameter values (typically when the binding and

unbinding rates are large) and hence should be used more cautiously. Ongoing work aims to extend the method to predict both mRNA and protein dynamics, including their joint distribution for pairs of genes.

Concluding, we have devised a method that overcomes many of the known difficulties encountered when simulating complex stochastic gene network dynamics. We anticipate that Holimap will be useful for investigating noisy dynamical phenomena in complex regulatory networks where intuitive understanding is challenging to attain and simulations using the SSA become computationally prohibitive.

## Methods
### Determining the effective parameter for the LMA
For the linear network in Fig. 2b, the evolution of moments is governed by

$$
\begin{aligned}
\dot{g}_0 &= \sigma_u g_1 - \hat{\sigma}_b g_0, \\
\dot{\mu}_{1,0} &= \rho_u B g_0 - d\mu_{1,0} + \sigma_u \mu_{1,1} - \hat{\sigma}_b \mu_{1,0}, \\
\dot{\mu}_{1,1} &= \rho_b B g_1 - d\mu_{1,1} - \sigma_u \mu_{1,1} + \hat{\sigma}_b \mu_{1,0}.
\end{aligned}
\tag{5}
$$

Inserting Eq. (2) into Eq. (5) gives a closed set of moment equations, from which the values of $g_0$, $\mu_{1,1}$, and $\mu_{1,0}$ can be computed approximately. Finally, using these values, the effective parameter $\hat{\sigma}_b$ can be obtained from Eq. (2). The remaining steps for the LMA are the same as for the 2-HM.

### Determining the effective parameters for the 4-HM
For the autoregulatory circuit, it follows from Eq. (1) that

$$
\begin{aligned}
\dot{\mu}_{1,0} + \dot{\mu}_{1,1} &= \rho_u B g_0 + \rho_b B g_1 - d(\mu_{1,0} + \mu_{1,1}) \\
&\quad + \sigma_u g_1 - \sigma_b \mu_{1,0}, \\
\dot{\mu}_{2,0} + \dot{\mu}_{2,1} &= 2\rho_u B(\mu_{1,0} + Bg_0) + 2\rho_b B(\mu_{1,1} + Bg_1) \\
&\quad - 2d(\mu_{2,0} + \mu_{2,1}) + 2\sigma_u \mu_{1,1} - 2\sigma_b \mu_{2,0}.
\end{aligned}
\tag{6}
$$

For the linear network in Fig. 2d, the evolution of moments is governed by

$$
\begin{aligned}
\dot{g}_0 &= \bar{\sigma}_u g_1 - \bar{\sigma}_b g_0, \\
\dot{\mu}_{1,0} &= \bar{\rho}_u B g_0 - d\mu_{1,0} + \bar{\sigma}_u \mu_{1,1} - \bar{\sigma}_b \mu_{1,0}, \\
\dot{\mu}_{1,1} &= \bar{\rho}_b B g_1 - d\mu_{1,1} - \bar{\sigma}_u \mu_{1,1} + \bar{\sigma}_b \mu_{1,0}, \\
\dot{\mu}_{2,0} &= 2\bar{\rho}_u B(\mu_{1,0} + Bg_0) - 2d\mu_{2,0} + \bar{\sigma}_u \mu_{2,1} - \bar{\sigma}_b \mu_{2,0}, \\
\dot{\mu}_{2,1} &= 2\bar{\rho}_b B(\mu_{1,1} + Bg_1) - 2d\mu_{2,1} - \bar{\sigma}_u \mu_{2,1} + \bar{\sigma}_b \mu_{2,0}.
\end{aligned}
\tag{7}
$$

From these equations, it is easy to show that

$$
\begin{aligned}
\dot{\mu}_{1,0} + \dot{\mu}_{1,1} &= \bar{\rho}_u B g_0 + \bar{\rho}_b B g_1 - d(\mu_{1,0} + \mu_{1,1}), \\
\dot{\mu}_{2,0} + \dot{\mu}_{2,1} &= 2\bar{\rho}_u B(\mu_{1,0} + Bg_0) + 2\bar{\rho}_b B(\mu_{1,1} + Bg_1) \\
&\quad - 2d(\mu_{2,0} + \mu_{2,1}).
\end{aligned}
\tag{8}
$$

Matching Eqs. (6) and (8), we find that $\bar{\rho}_b$ and $\bar{\rho}_u$ should satisfy the following system of linear equations:

$$
\begin{aligned}
\bar{\rho}_u B g_0 + \bar{\rho}_b B g_1 &= \rho_u B g_0 + \rho_b B g_1 + \sigma_u g_1 - \sigma_b \mu_{1,0}, \\
\bar{\rho}_u B(\mu_{1,0} + Bg_0) &+ \bar{\rho}_b B(\mu_{1,1} + Bg_1) \\
&= \rho_u B(\mu_{1,0} + Bg_0) + \rho_b B(\mu_{1,1} + Bg_1) + \sigma_u \mu_{1,1} - \sigma_b \mu_{2,0}.
\end{aligned}
\tag{9}
$$

Matching the first and third identities in Eqs. (1) and (7), it is clear that $\bar{\sigma}_b$ and $\bar{\sigma}_u$ should satisfy the following system of linear equations:

$$
\begin{aligned}
\bar{\sigma}_u g_1 - \bar{\sigma}_b g_0 &= \sigma_u g_1 - \sigma_b \mu_{1,0}, \\
\bar{\sigma}_u \mu_{1,1} - \bar{\sigma}_b \mu_{1,0} &= \sigma_u \mu_{1,1} - \sigma_b \mu_{2,0} + (\bar{\rho}_b - \rho_b)Bg_1,
\end{aligned}
\tag{10}
$$

where $\bar{\rho}_b$ has been determined by solving Eq. (9). Compared with Eq. (4), Eq. (10) has an additional term $(\bar{\rho}_b - \rho_b)Bg_1$. This is because $\rho_b$ remains unchanged for the 2-HM but is changed for the 4-HM.

Finally, inserting Eqs. (9) and (10) into Eq. (7) gives a system of closed moment equations and hence the values of all zeroth, first, and

second-order moments can be approximately calculated. Substituting these moments into Eqs. (9) and (10), one can finally solve for the four effective parameters $\bar{\rho}_u, \bar{\rho}_b, \bar{\sigma}_u,$ and $\bar{\sigma}_b$ of the linear network. The 4-HM predicts the protein distribution of the nonlinear network by solving the CME of the linear one in Fig. 2d with the values of the effective parameters found above.

## Stochastic model for complex gene networks

Here we consider the stochastic model of an arbitrary gene regulatory network involving protein synthesis, protein degradation, gene state switching, and complex gene regulation mechanisms[63]. Specifically, we assume that the network involves $M$ distinct genes, each of which can be in two states: an inactive state $G_j$ and an active state $G_j^*$. The protein associated with gene $G_j$ is denoted by $P_j$. The network can be described by the following reactions:

$$
\begin{aligned}
G_j \xrightarrow{\alpha_j^0} G_j^*, \quad & G_j^* \xrightarrow{\alpha_j^1} G_j, \\
G_j + h_{ji}P_i \xrightarrow{\sigma_{ji}^0} G_j^*, \quad & G_j^* + h_{ji}P_i \xrightarrow{\sigma_{ji}^1} G_j, \\
G_j \xrightarrow{\rho_j^0} G_j + P_j, \quad & G_j^* \xrightarrow{\rho_j^1} G_j^* + P_j, \\
P_j \xrightarrow{d_j} \varnothing, \quad & i,j = 1,2,\ldots,M,
\end{aligned}
\tag{11}
$$

where the reactions in the first row describe spontaneous gene state switching, the reactions in the second row describe gene regulation, the reactions in the third row describe protein synthesis in the two-gene states, and the last reaction describes protein degradation or dilution. Since $G_j$ is the inactive state and $G_j^*$ is the active state, we have $\rho_j^1 > \rho_j^0$. Due to complex gene regulation, each gene $G_j$ may be regulated by all genes. If gene $G_i$ activates gene $G_j$, then $\sigma_{ji}^0 > 0$ and $\sigma_{ji}^1 = 0$ since the binding of protein $P_i$ induces the switching from $G_j$ to $G_j^*$; on the contrary, if gene $G_i$ inhibits gene $G_j$, then $\sigma_{ji}^0 = 0$ and $\sigma_{ji}^1 > 0$ since the binding of protein $P_i$ induces the switching from $G_j^*$ to $G_j$. When performing simulations (SSA and SSA + Holimap), the parameters are chosen as $d_i = 1, h_{ji} = 1, \rho_j^1 = 81, \rho_j^0 = 5.4, \alpha_j^0 = \alpha_j^1 = 0.5, \sigma_{ji}^0 = 0.01, \sigma_{ji}^1 = 0$ when $G_i$ activates $G_j$, and $\sigma_{ji}^0 = 0, \sigma_{ji}^1 = 0.01$ when $G_i$ inhibits $G_j$. The presence or absence of a gene-gene interaction and its type are determined randomly. Here we assume that protein-gene interactions are non-cooperative ($h_{ij} = 1$). The spontaneous switching rates between $G_j$ and $G_j^*$ are chosen to be $\sigma_j^0 = \sigma_j^1 = 0.5$. Since each gene is on average regulated by two genes (one positive regulation and one negative regulation), the switching rates due to gene regulation are roughly equal to $\sigma_{ji}^0 = \sigma_{ji}^1 = 0.01$ multiplied by the number of regulator $P_i$, which is ~50. Hence the total switching rates due to spontaneous contributions and gene regulation are roughly $0.5 + 0.01 \times 50 = 1$, i.e., they are comparable with the degradation rate $d_i = 1$.

## Reporting summary

Further information on research design is available in the Nature Portfolio Reporting Summary linked to this article.

## Data availability

MATLAB R2019a was used to analyze the data. Source data are provided with this paper.

## Code availability

The MATLAB codes for Holimap and SSA + Holimap can be found in the Github repository[64].

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

## Acknowledgements

We thank Augustinas Sukys for comments on the manuscript. C.J. acknowledges support from the National Natural Science Foundation of China with grant Nos. U2230402 and 12271020. R.G. acknowledges support from the Leverhulme Trust (RPG-2020-327).

## Author contributions

R.G. conceived the original idea. C.J. performed the theoretical derivations and numerical simulations. C.J. and R.G. interpreted the theoretical results and jointly wrote the manuscript.

## Competing interests

The authors declare no competing interests.
