## [Peer Review File · Nature Communications]

Holimap: an accurate and efficient method for solving stochastic gene network dynamicsReviewer #1 (Remarks to the Author):

Comments: In this study, the authors introduced a novel computational framework termed Holimap, which is designed to enhance the precision and effectiveness in the modeling of stochastic gene network dynamics. By conducting a comparative analysis with established methodologies such as the Stochastic Simulation Algorithm (SSA) and Finite State Projection (FSP), the authors elucidated the computational advantages conferred by Holimap. Furthermore, they proposed a hybrid strategy that integrates SSA with Holimap, demonstrating a significant reduction in computational time while maintaining accuracy across a variety of gene regulatory networks. Despite the technical robustness of this paper and its contribution to the advancement of the field, opportunities exist to augment its accessibility and impact amongst a broader audience, notably experimental biologists and individuals lacking specialization in computational modeling of gene regulatory networks. Suggestions for Improving Paper Accessibility:

- To enhance the manuscript accessibility, a more comprehensive introduction to the Stochastic Simulation Algorithm (SSA) and Finite State Projection (FSP) techniques would be highly advantageous. Including a concise historical overview, outlining the fundamental principles of these approaches, and highlighting their typical uses could aid biologists lacking computational expertise in grasping the value of the advancements brought by Holimap. Additionally, incorporating visual aids or straightforward examples to demonstrate the workings of these methods would further improve understanding.
- It is advisable to incorporate a roadmap of the paper at the conclusion of the introduction. This roadmap should outline the main sections of the document, provide a brief overview of the methodology employed by Holimap, its application to various gene regulatory networks, and a summary of key findings and their implications. This would enable readers of varying expertise to swiftly grasp the structure and substance of the paper.
- A demonstration of Holimap's application, particularly the SSA+Holimap hybrid approach, to real biological or experimental data would enhance the manuscript. Detailing specific case studies where Holimap has been utilized to analyze experimental data or predict outcomes of biological experiments would showcase its practical utility. Highlighting instances where Holimap has unveiled new biological insights or supported experimental design would emphasize its relevance to experimental biologists.
- Guidance for experimentalists/biologists on leveraging Holimap in their research could prove invaluable. This might encompass a straightforward tutorial or a step-by-step guide for applying the hybrid approach to analyze gene expression data, predict outcomes of gene knockouts, or model the effects of environmental changes on gene regulatory networks. Sharing links to user-friendly software interfaces or online platforms that facilitate Holimap use and offer comprehensive tutorials would be beneficial.

Here are a list of questions we have that need to be clarified.

(1) The authors introduced the higher order moments of the protein numbers. It might be worthwhile to justify the definition and why they are important, like being the measurement of protein-protein interaction. The authors discussed 2nd moments in the paper. Could the authors use the third moments or higher order moments in the computation or discussion?

(2) Does the Holimap approach require the running of the other models like LMA, or SSA initially to determine the values of the parameters? Could it be done independent of other computational models?

(3) at the very end of page 10, the authors wrote "while performing simulations, the parameters are chosen as.....". Could the authors elaborate a little more on what

simulation is used and how the parameters are chosen?

(4) In the hybrid SSA + Holimap approach, the authors do not provide details on how the number of SSA trajectories (N) is determined. Could the authors elaborate on the criteria or algorithm used to choose an appropriate value of N, particularly for larger networks?

(5) In the "Stochastic model for complex gene networks" section, the authors note that the computational expense of Holimap is influenced by both the intricacy of the regulatory network and the quantity of moment equations (L) that need to be addressed. Could the authors offer a more precise evaluation or description of how the computational cost scales in relation to the network size (number of genes) and the level of cooperativity?

(6) The paper centers on gene regulatory networks that involve particular regulatory mechanisms like cooperative binding, autoregulation, toggle switch, and repressilator. Could the authors discuss the potential constraints or difficulties in implementing Holimap for different kinds of regulatory mechanisms, such as post-transcriptional regulation or intricate gene-gene interactions?

The introduction of Holimap represents a notable advancement in computational techniques for the analysis of gene regulatory networks. We suggest the authors implement the aforementioned recommendations and clarify all questions listed to enhance the manuscript's readability and highlight Holimap's practical application to a wider audience

Reviewer #2 (Remarks to the Author):

Chemical master equations are one of the most effective ways to formulate the dynamics of a chemical reaction especially under prevalently stochastic conditions. However, the infinite hierarchy of the time evolution equations of probability distribution limits the utilization of this framework. In this paper, the authors have attempted to mitigate this limitation in a group of reaction networks. The authors developed a method called Holimap that maps the $n+h$ th order central moments (h =cooperativity) to n th order central moments by approximation of 2 parameters (2HM) or 4 parameters (4HM). Both approaches yield significant improvement over the conventional Linear Mapping Approximation (LMA) method. Matching the typical trend, the higher order approximation, i.e. 4HM, is less numerically stable than the lower order 2HM. Hence, for large networks the authors utilized a hybrid method of approximation (SSA+2HM) which turned out to be computationally efficient in predicting the protein distribution. While the improvement of speed and accuracy is encouraging, the authors' extension of LMA seems to be useful in a very specific type of gene interaction networks (i.e. transcriptional networks involving protein-DNA interactions whose impact on protein concentrations is negligible). As it stands, the work does not seem to be broadly useful to study 'the intricate network of gene-gene interactions' as claimed by the authors. In addition, there seems to be a lack of comparison between experimental measurements and the predictions from the models or the parameter region investigated for method performance. My comments are suggestions are as follows.

1. Limited gene regulatory circuits described by linear-mapping-based methods. With the replacement of $G+hP \leftrightarrow G^*$ by $G \leftrightarrow G^*$, Holimap and other linear-mapping-based methods seem to assume that an elementary chemical reaction consumes no more than one reactant with a linear rate. This assumption is reasonable in the scenario that the amount of regulator is much greater than that of the target. This approach is therefore valid in a special system where a transcription factor binds to one or a few promoters, as shown in the paper. However, the assumption can be problematic in many realistic scenarios where gene regulation involves regulator-target reciprocity in a single reaction. For example, in post-translational gene regulation, sequestrations of enzymes involved in consecutive steps of reactions can produce nontrivial protein distributions underlined by multistability and oscillation (10.1529/biophysj.105.073874, <https://doi.org/10.1091/mbc.E16-03-0137>). In post-transcriptional gene regulation, microRNA and its mRNA target can interact at comparable concentrations, which can also generate emerging gene expression dynamics (<https://doi.org/10.1093/nar/gkac217>). Even for transcriptional control studied in this paper, there is an increasing recognition of multiple DNA sites compete for or sequester regulatory factors

(<https://doi.org/10.1242/dev.201989>). It is unclear how the proposed approach can be applied to any of these important types of gene-gene interactions and gene regulations.

2. Biological relevance of the parameter region $\sigma \geq d$. The authors showed that Holimap outperforms canonical LMA mainly in the parameter region $\sigma \geq d$. However, there is no assessment on the biological importance of this parameter region. This is a missed opportunity to relate this work to experimental observations. I suggest that the authors convert their dimensionless models (back) to those with physical quantities including time, and demonstrate the importance of parameter region $\sigma \geq d$ by comparing it with experimental measurements.

3. Comparison with neural network-based approaches. The implementation of SSA+2HM is akin to approximating solutions with neural networks (e.g. <https://doi.org/10.1016/j.isci.2022.105010> from the authors' group) in that solutions from SSA are used to train more efficient surrogate models. This raises a question of what advantages the SSA+2HM method has over neural networks in terms of speed and accuracy. I suggest that the authors make a systematic comparison between the two approaches.

4. Numerical Stability Assessment. The numerical stability is one of the most concerning factors for any moment closure approximation designed to study realistic networks. The authors have mentioned that 4HM is less numerically stable than 2HM. However, I suggest the authors carry out a systematic investigation which quantifies the numerical stability of the processes (4HM, 2HM, SSA+2HM).

5. Assumptions and limitations of moment closure approach. It would also be beneficial for the readers if the authors can clearly describe the assumptions and limitations of the Holimap's moment closure approach either in the main text or in the supplementary text.

6. Clarification of Holimap algorithm. The process of parameter approximation is the pivotal point of the Holimap algorithm. The current version of the manuscript lacks clarity in explaining the intricacies of the algorithm. Perhaps a flow chart in the supplementary document or in the GitHub repository will enhance the reproducibility and lucidity of the work. (the authors should emphasize the process of approximating the parameters in the flow chart at least for 2HM).

7. Typo Correction: Main text (Page1, Column2, Line1): efficiently.

8. Typo Correction: Supplementary (Equation 6): $\bar{}$ in the third line.

Reviewer #2 (Remarks on code availability):

As indicated in the comments, an illustration of the algorithm in the context of the code files will be helpful to readers.

Furthermore, extension to networks beyond autoregulation and repressilator (such as the 'random network' in the paper) is missing from the repo.

Reviewer #3 (Remarks to the Author):

Reviewer #4 (Remarks to the Author):

The paper under review presents a novel method, Holimap, for approximating gene regulatory networks with non-linear interactions (reaction rates) by simpler networks with linear reaction rates. The method is applied to several examples, including autoregulatory feedback loops and two- and three-gene networks that exhibit mono- and bistability or oscillatory behaviour. These

examples illustrate that the method can faithfully capture steady-state and time-dependent distributions of protein numbers as well as that it outperforms LMA, a previously reported approximation method. Finally, the authors propose a hybrid method that combines stochastic simulations with Holimap, for obtaining smooth distributions from complex gene networks with high efficiency.

The study is of interest as it provides with a useful tool for studying stochastic gene networks for which obtaining analytical probabilistic solutions is impossible or computationally expensive. Moreover, I believe that it will motivate further studies that will develop alternative approximation methods. Overall, I found the paper to be well written and the method is carefully explained and illustrated in detail. I have the following few comments:

- Even though the description of the method through the various examples and in the SI is mostly clear, I think it would be useful for the overall presentation if the authors provide a box or a diagram that describes the different steps of the Holimap algorithm in a more generic way. This could be done for an arbitrary gene network (e.g. the network shown in the last paragraph of Methods).

- I noticed that in the autoregulation example, where the 2-HM method is presented for the first time, the authors match the zeroth- and first-order moment equations of the non-linear and the corresponding linear system to obtain the effective gene switching parameters. However, when describing the method and the same example in the SI, they mention that the effective parameters are chosen such that all moments up to second order of the two systems are matched. It is true that the zeroth- and first-order moment equations depend also on the 2nd moment of the gene in the unbound state, however there is no dependence on the other 2nd moment. I found this detail a bit confusing and perhaps the authors could explain this better in the main text.

- In the same part of the Results, the authors explain how to obtain the effective parameters when the system has not reached steady state. I found that this part was not as clearly explained, so the authors could elaborate a bit more (perhaps in the SI).

- In several parts of the paper, the authors comment on whether the 2-Holimap or the 4-Holimap is a better choice of approximation. I think it would be useful if the authors made a short summary / table that outlines in what types of scenarios one should choose to use 2-HM over 4-HM or vice versa. On a similar note, I found the hybrid method combining SSA and Holimap interesting and it would be useful to summarise when the hybrid approach should be preferred over a standard Holimap approach. Such a summary would help to better highlight the versatility of Holimap and its hybrid variants in analysing gene networks of different complexities.

- It is suggested that the authors provide documentation in their Github repository, so that the Holimap method is reproducible.

Reviewer #4 (Remarks on code availability):

There is no README provided, but it has been suggested to the authors to provide documentation so their method is reproducible.

Reviewer #5 (Remarks to the Author):

Response to Referee 1

Thank you for your valuable comments and suggestions which have helped us greatly in improving the quality of this paper. Below we provide a point-by-point response to each comment. All modifications made in the revised manuscript are shown in blue.

0. In this study, the authors introduced a novel computational framework termed Holimap, which is designed to enhance the precision and effectiveness in the modeling of stochastic gene network dynamics. By conducting a comparative analysis with established methodologies such as the Stochastic Simulation Algorithm (SSA) and Finite State Projection (FSP), the authors elucidated the computational advantages conferred by Holimap. Furthermore, they proposed a hybrid strategy that integrates SSA with Holimap, demonstrating a significant reduction in computational time while maintaining accuracy across a variety of gene regulatory networks. Despite the technical robustness of this paper and its contribution to the advancement of the field, opportunities exist to augment its accessibility and impact amongst a broader audience, notably experimental biologists and individuals lacking specialization in computational modeling of gene regulatory networks.

Response: Thank you for the positive comments and valuable suggestions. In the revised manuscript, we have tried to improve the accessibility and impact of the paper amongst a broader audience. The detailed modifications are described as follows.

Suggestions for Improving Paper Accessibility

1. To enhance the manuscript accessibility, a more comprehensive introduction to the Stochastic Simulation Algorithm (SSA) and Finite State Projection (FSP) techniques would be highly advantageous. Including a concise historical overview, outlining the fundamental principles of these approaches, and highlighting their typical uses could aid biologists lacking computational expertise in grasping the value of the advancements brought by Holimap. Additionally, incorporating visual aids or straightforward examples to demonstrate the workings of these methods would further improve understanding.

Response: In the revised manuscript, we have made a more comprehensive introduction to the SSA and FSP techniques (see page 1). Specifically, we have clarified their fundamental principles and highlighted their typical uses. We also added some references that refers the reader to a variety of resources where more information is available about simulation methods in stochastic biology.

In addition, we also tried to incorporate some visual aids to demonstrate the workings of the SSA and FSP — the dices and trajectories in the upper row of Figure 1 indicates that the SSA simulates the waiting time and state change of every single reaction and generates a large number of statistically correct trajectories from which the copy number distributions of all biochemical species can be obtained; the lattices in the lower row of Figure 1 indicate that FSP truncates an infinite state space into a finite one and then solves the finite-dimensional CME. The caption of Figure 1 has been updated to clarify the differences between the SSA and FSP.

2. It is advisable to incorporate a roadmap of the paper at the conclusion of the introduction. This roadmap should outline the main sections of the document, provide a brief overview of the methodology employed by Holimap, its

application to various gene regulatory networks, and a summary of key findings and their implications. This would enable readers of varying expertise to swiftly grasp the structure and substance of the paper.

Response: In the revised manuscript, we added a new paragraph at the end of the Introduction to provide an overview of the paper (see page 2).

3. A demonstration of Holimap's application, particularly the SSA+Holimap hybrid approach, to real biological or experimental data would enhance the manuscript. Detailing specific case studies where Holimap has been utilized to analyze experimental data or predict outcomes of biological experiments would showcase its practical utility. Highlighting instances where Holimap has unveiled new biological insights or supported experimental design would emphasize its relevance to experimental biologists.

Response: Thank you for your valuable suggestion. To clarify that the parameter ranges we used in the paper are biologically relevant, we have added a new paragraph to explicitly show that this is the case using experimental data for mammalian cells (see page 5). Direct application of Holimap to experimental data is our eventual goal but not in this paper. In the current paper, we have just enough space to introduce the technical aspects of the method and thoroughly verify its application to a large variety of gene regulatory systems (as we mention later, we have now shown Holimap's application to post-translational and post-transcriptional networks which occupies almost two new pages). The application of Holimap to experimental data is a large project in itself because the parameters of these networks are typically unknown and hence these have to be first estimated before the dynamics can be explored. We currently devising a Holimap-based parameter and network inference method which we hope to report on in a forthcoming paper.

4. Guidance for experimentalists/biologists on leveraging Holimap in their research could prove invaluable. This might encompass a straightforward tutorial or a step-by-step guide for applying the hybrid approach to analyze gene expression data, predict outcomes of gene knockouts, or model the effects of environmental changes on gene regulatory networks. Sharing links to user-friendly software interfaces or online platforms that facilitate Holimap use and offer comprehensive tutorials would be beneficial.

Response: Thank you for your valuable suggestion. In the revised manuscript, we have added a flow chart that encompasses a step-by-step guide for applying the hybrid SSA+Holimap approach to various gene networks (see page 4 and Supplementary Figure 1). However, as in the response to the previous comment, direct application to experimental data is our eventual goal but we cannot possibly accomplish it within the space available for the current paper. A new method needs to be thoroughly tested before being made available for general use and our current paper accomplishes this. In the future, we plan to extend an existing software package developed by the Grima group, MomentClosure.jl (<https://academic.oup.com/bioinformatics/article/38/1/289/6309452>), to incorporate Holimap. As we did for the LMA (the forerunner of Holimap), we will provide a step-by-step tutorial to its use but probably only online. However, we realize that this is a project by itself, requiring substantial code writing in Julia to make it both user-friendly and applicable to a vast range of regulatory networks. Hence we hope to provide this towards the end of this year or beginning of next year. We also note that in the meanwhile, others can implement Holimap for their own applications by modifying the Matlab code for all the networks studied in the paper which is available at <https://github.com/chenjiacsrc/Holimap>.

Technical comments

(1) *The authors introduced the higher order moments of the protein numbers. It might be worthwhile to justify the definition and why they are important, like being the measurement of protein-protein interaction. The authors discussed 2nd moments in the paper. Could the authors use the third moments or higher order moments in the computation or discussion?*

Response: In the revised manuscript, we emphasized that higher-order moments need to be used when the protein-gene interactions are cooperative ($h \geq 2$). In particular, when $h = 2$, third-order moment equations need to be solved and the effective parameters depend on the values of zeroth, first, second, and third-order moments (see page 4).

(2) *Does the Holimap approach require the running of the other models like LMA, or SSA initially to determine the values of the parameters? Could it be done independent of other computational models?*

Response: In the revised manuscript, we emphasized that for (standard) Holimap, the determination of the effective parameters can be done independent of other computational methods. However, the hybrid SSA+Holimap method requires the running of the SSA to determine the effective parameters (see page 8).

(3) *at the very end of page 10, the authors wrote “while performing simulations, the parameters are chosen as.....”. Could the authors elaborate a little more on what simulation is used and how the parameters are chosen?*

Response: In the revised manuscript, we emphasized that both the SSA and SSA+Holimap are used here. Moreover, the parameters are chosen based on the following considerations. Here, for simplicity, we assume that protein-gene interactions are non-cooperative ($h_{ij} = 1$). The spontaneous switching rates between G_j and G_j^* are chosen to be $\alpha_j^0 = \alpha_j^1 = 0.5$. Since each gene is on average regulated by two genes (one positive regulation and one negative regulation), the switching rates due to gene regulation are roughly equal to $\sigma_{ji}^0 = \sigma_{ji}^1 = 0.01$ multiplied by the number of regulator P_i , which is about 50. Hence the total switching rates due to spontaneous contributions and gene regulation are roughly $0.5 + 0.01 \times 50 = 1$, i.e. they are comparable with the degradation rate $d_i = 1$ (see page 13).

(4) *In the hybrid SSA + Holimap approach, the authors do not provide details on how the number of SSA trajectories (N) is determined. Could the authors elaborate on the criteria or algorithm used to choose an appropriate value of N , particularly for larger networks?*

Response: In the revised manuscript, we elaborated on the criterion used to determine the number of SSA trajectories. In fact, the number of SSA trajectories N needed for the hybrid SSA+Holimap approach is chosen such that the molecule number distributions obtained from N trajectories and those obtained from $3N$ trajectories have an HD (averaged over all time points) less than 0.02, i.e. increasing the sample size will not substantially improve the approximation accuracy. We find that for most networks, a sample size of $N = 2000$ is sufficient to

satisfy this criterion (see page 9).

(5) In the “Stochastic model for complex gene networks” section, the authors note that the computational expense of Holimap is influenced by both the intricacy of the regulatory network and the quantity of moment equations (L) that need to be addressed. Could the authors offer a more precise evaluation or description of how the computational cost scales in relation to the network size (number of genes) and the level of cooperativity?

Response: The computational cost of Holimap is mainly determined by the number of moment equations, L , to be solved. In the revised manuscript, we proved that for a general network, L scales polynomially with the cooperativity h and scales exponentially with respect to the network size M , i.e. the number of genes (see page 4 and Supplementary Note 3).

(6) The paper centers on gene regulatory networks that involve particular regulatory mechanisms like cooperative binding, autoregulation, toggle switch, and repressilator. Could the authors discuss the potential constraints or difficulties in implementing Holimap for different kinds of regulatory mechanisms, such as post-transcriptional regulation or intricate gene-gene interactions?

Response: The previous version of the manuscript mainly focused on transcriptional networks with protein-gene interactions. In the revised manuscript, we show that Holimap can also be applied to solve the dynamics of post-transcriptional and post-translational networks with complex protein-protein, protein-RNA, and RNA-RNA interactions. For transcriptional networks, Holimap does not change the degradation rate; however, for post-transcriptional and post-translational networks, both the binding/unbinding rate and degradation rate need to be modified (see page 10, Figure 6, and Supplementary Note 7).

Response to Referee 2

Thank you for your valuable comments and suggestions which have helped us greatly in improving the quality of this paper. Below we provide a point-by-point response to each comment. All modifications made in the revised manuscript are shown in blue.

0. *Chemical master equations are one of the most effective ways to formulate the dynamics of a chemical reaction especially under prevalently stochastic conditions. However, the infinite hierarchy of the time evolution equations of probability distribution limits the utilization of this framework. In this paper, the authors have attempted to mitigate this limitation in a group of reaction networks. The authors developed a method called Holimap that maps the $(n + h)$ th order central moments (h =cooperativity) to n th order central moments by approximation of 2 parameters (2HM) or 4 parameters (4HM). Both approaches yield significant improvement over the conventional Linear Mapping Approximation (LMA) method. Matching the typical trend, the higher order approximation, i.e. 4HM, is less numerically stable than the lower order 2HM. Hence, for large networks the authors utilized a hybrid method of approximation (SSA+2HM) which turned out to be computationally efficient in predicting the protein distribution. While the improvement of speed and accuracy is encouraging, the authors' extension of LMA seems to be useful in a very specific type of gene interaction networks (i.e. transcriptional networks involving protein-DNA interactions whose impact on protein concentrations is negligible). As it stands, the work does not seem to be broadly useful to study 'the intricate network of gene-gene interactions' as claimed by the authors. In addition, there seems to be a lack of comparison between experimental measurements and the predictions from the models or the parameter region investigated for method performance. My comments are suggestions are as follows.*

Response: Thank you for your valuable comments and suggestions. Following your suggestions, in the revised manuscript, we have generalized Holimap to post-transcriptional and post-translational networks with complex protein-protein, protein-RNA, and RNA-RNA interactions. In addition, we have made a detailed comparison between experimental measurements and the parameter region investigated for method performance. The detailed modifications are described below.

1. *Limited gene regulatory circuits described by linear-mapping-based methods. With the replacement of $G + hP \rightleftharpoons G^*$ by $G \rightleftharpoons G^*$, Holimap and other linear-mapping-based methods seem to assume that an elementary chemical reaction consumes no more than one reactant with a linear rate. This assumption is reasonable in the scenario that the amount of regulator is much greater than that of the target. This approach is therefore valid in a special system where a transcription factor binds to one or a few promoters, as shown in the paper. However, the assumption can be problematic in many realistic scenarios where gene regulation involves regulator-target reciprocity in a single reaction. For example, in post-translational gene regulation, sequestrations of enzymes involved in consecutive steps of reactions can produce nontrivial protein distributions underlined by multistability and oscillation ([10.1529/biophysj.105.073874](https://doi.org/10.1529/biophysj.105.073874), <https://doi.org/10.1091/mbc.E16-03-0137>). In post-transcriptional gene regulation, microRNA and its mRNA target can interact at comparable concentrations, which can also generate emerging gene expression dynamics (<https://doi.org/10.1093/nar/gkac217>). Even for transcriptional control studied in this paper, there is an increasing recognition of multiple DNA sites compete for or sequester regulatory factors (<https://doi.org/10.1242/dev.201989>). It is unclear how the proposed approach can be applied to any of these important types of gene-gene interactions and gene regulations.*

Response: The previous version of the manuscript mainly focused on transcriptional networks with protein-gene interactions. In the revised manuscript, we showed that Holimap can also be applied to solve the dynamics of post-transcriptional and post-translational networks with complex protein-protein, protein-RNA, and RNA-RNA interactions. For transcriptional networks, Holimap does not change the degradation rate; however, for post-transcriptional and post-translational networks, both the binding/unbinding rate and degradation rate need to be modified (see page 10, Figure 6, and Supplementary Note 7).

In Supplementary Note 5, we also applied Holimap to two-node transcriptional networks with two TF binding sites in the promoter region. We showed that Holimap performs excellently in capturing the protein distributions as well as the bimodal region, independent of the type of network topology and the type of TF binding (independent, positive cooperative, and negative cooperative binding). Intriguingly, we found that positive (negative) cooperativity of the two TF binding sites enhances bimodality for networks with coherent (incoherent) inputs (see page 7 and Supplementary Note 5).

Recall that Holimap replaces the second-order interaction $A + B \rightarrow C$ by the effective first-order reaction $A \rightarrow C$ with a new parameter. Intuitively, this approximation is valid when molecule B is very abundant compared to molecule A so that noise in molecule B number can be ignored. However, we find that Holimap makes accurate predictions not only in this special case, but also in the scenarios where the two molecules interact at comparable concentrations and where B is very scarce compared to A . This again confirms the high accuracy of Holimap over a wide range of parameter values (see page 10 and Supplementary Figure 5).

2. Biological relevance of the parameter region $\sigma_u \geq d$. The authors showed that Holimap outperforms canonical LMA mainly in the parameter region $\sigma_u \geq d$. However, there is no assessment on the biological importance of this parameter region. This is a missed opportunity to relate this work to experimental observations. I suggest that the authors convert their dimensionless models (back) to those with physical quantities including time, and demonstrate the importance of parameter region $\sigma_u \geq d$ by comparing it with experimental measurements.

Response: In the revised manuscript, we converted the dimensionless model back to the original model including time. Moreover, we have demonstrated the importance of the parameter region $\sigma_u \geq d$ by comparing it with experimental measurements in mouse fibroblasts (see page 5).

3. Comparison with neural network-based approaches. The implementation of SSA+2HM is akin to approximating solutions with neural networks (e.g. <https://doi.org/10.1016/j.isci.2022.105010> from the authors' group) in that solutions from SSA are used to train more efficient surrogate models. This raises a question of what advantages the SSA+2HM method has over neural networks in terms of speed and accuracy. I suggest that the authors make a systematic comparison between the two approaches.

Response: In the revised manuscript, we have discussed the differences between the hybrid SSA+Holimap method and neural network-based approaches (see page 12). Indeed, our hybrid SSA+Holimap method shares some similarities with neural network-based approaches, which can also be used to predict complex gene network dynamics. The former uses the SSA to generate the sample moments which are then used to compute the values

of effective parameters, while the latter uses the SSA to train the surrogate neural network model. While both methods can accurately capture the protein/mRNA distribution, our method outperforms the neural network-based ones in three aspects: (i) while neural network models perform well in the parameter ranges which are used to train the surrogate model, their extrapolation ability is usually weak. Our method is mechanism-based and provides accurate results over wide parameter ranges; (ii) neural network-based methods require a very long time to train the surrogate model. When the network is complex, the training time may take tens of hours to several days and may also require multiple rounds of hyperparameter tuning. In contrast, Holimap avoids the long training time; (iii) neural network models have good predictive ability but their learnt approximation does not typically have a clear biophysical interpretation. Holimap transforms the complex nonlinear network into a linear one which not only has a clear physical meaning but also allows an approximative analytical solution. In addition, SSA+Holimap can be combined with neural network-based methods to increase the speed and accuracy of the latter. Since SSA+Holimap can be used to generate distributions comparable in accuracy to those from the SSA with a much larger number of trajectories, it follows that SSA+Holimap can reduce the time to generate an accurate training dataset as input to the neural network.

4. Numerical Stability Assessment. The numerical stability is one of the most concerning factors for any moment closure approximation designed to study realistic networks. The authors have mentioned that 4HM is less numerically stable than 2HM. However, I suggest the authors carry out a systematic investigation which quantifies the numerical stability of the processes (4HM, 2HM, SSA+2HM).

Response: In the revised manuscript, we quantified the numerical stability of the two types of Holimaps (2-HM and 4-HM) for autoregulatory feedback loops. We emphasized that depending on parameter values, the 4-HM may lead to numerical instability at short times, which usually occurs when σ_u and σ_b are large for negative feedback loops. However, we did not observe numerical instability for the 2-HM (and also for the hybrid SSA+2-HM) (see page 5 and Supplementary Figure 3).

5. Assumptions and limitations of moment closure approach. It would also be beneficial for the readers if the authors can clearly describe the assumptions and limitations of the Holimap's moment closure approach either in the main text or in the supplementary text.

Response: In the revised manuscript, we clarified the assumptions and limitations of Holimap's moment closure approach (see page 5 in Supplementary Information). We emphasized that our moment closure method essentially assumes that the marginal dynamics of a complex nonlinear network can be approximated by that of a decoupled linear network and it provides a way to find the optimal approximation. The advantage of our moment closure method is that it does not assume in advance the shape of the protein distribution; specifically, it does not assume the protein distribution to be of a simple type such as the Gaussian, Poisson, Log-normal, or Gamma distributions, as commonly assumed by conventional moment closure methods. As stated in the main text, the 4-HM may lead to numerical instability when computing the time-dependent distributions. This is the main limitation of our moment closure technique.

6. Clarification of Holimap algorithm. The process of parameter approximation is the pivotal point of the Holimap algorithm. The current version of the manuscript lacks clarity in explaining the intricacies of the algorithm.

Perhaps a flow chart in the supplementary document or in the GitHub repository will enhance the reproducibility and lucidity of the work. (the authors should emphasize the process of approximating the parameters in the flow chart at least for 2HM).

Response: In the revised manuscript, we have added a flow chart that describes the main steps of the Holimap algorithm (see page 4 and Supplementary Figure 1).

7. *Typo Correction: Main text (Page1, Column2, Line1): efficiently.*

Response: Corrected as suggested.

8. *Typo Correction: Supplementary (Equation 6): $\bar{\cdot}$ in the third line.*

Response: Corrected as suggested.

Remarks on code availability

1. *As indicated in the comments, an illustration of the algorithm in the context of the code files will be helpful to readers. Furthermore, extension to networks beyond autoregulation and repressilator (such as the 'random network' in the paper) is missing from the repo.*

Response: In the revised manuscript, we have now provided the MATLAB codes for a random eight-node network, two post-translational networks, and two post-transcriptional networks. Moreover we have also added a README file in the GitHub repository.

Response to Referee 3

Thank you for your valuable comments and suggestions which have helped us greatly in improving the quality of this paper. Below we provide a point-by-point response to each comment. All modifications made in the revised manuscript are shown in blue.

1. *I co-reviewed this manuscript with one of the reviewers who provided the listed reports. This is part of the Nature Communications initiative to facilitate training in peer review and to provide appropriate recognition for Early Career Researchers who co-review manuscripts.*

Response: Thank you for your valuable comments and suggestions.

Response to Referee 4

Thank you for your valuable comments and suggestions which have helped us greatly in improving the quality of this paper. Below we provide a point-by-point response to each comment. All modifications made in the revised manuscript are shown in blue.

0. *The paper under review presents a novel method, Holimap, for approximating gene regulatory networks with non-linear interactions (reaction rates) by simpler networks with linear reaction rates. The method is applied to several examples, including autoregulatory feedback loops and two- and three-gene networks that exhibit mono- and bistability or oscillatory behaviour. These examples illustrate that the method can faithfully capture steady-state and time-dependent distributions of protein numbers as well as that it outperforms LMA, a previously reported approximation method. Finally, the authors propose a hybrid method that combines stochastic simulations with Holimap, for obtaining smooth distributions from complex gene networks with high efficiency.*

The study is of interest as it provides with a useful tool for studying stochastic gene networks for which obtaining analytical probabilistic solutions is impossible or computationally expensive. Moreover, I believe that it will motivate further studies that will develop alternative approximation methods. Overall, I found the paper to be well written and the method is carefully explained and illustrated in detail. I have the following few comments:

Response: Thank you for the positive comments.

1. *Even though the description of the method through the various examples and in the SI is mostly clear, I think it would be useful for the overall presentation if the authors provide a box or a diagram that describes the different steps of the Holimap algorithm in a more generic way. This could be done for an arbitrary gene network (e.g. the network shown in the last paragraph of Methods).*

Response: In the revised manuscript, we have added a flow chart that describes the main steps of the Holimap algorithm (see page 4 and Supplementary Figure 1).

2. *I noticed that in the autoregulation example, where the 2-HM method is presented for the first time, the authors match the zeroth- and first-order moment equations of the non-linear and the corresponding linear system to obtain the effective gene switching parameters. However, when describing the method and the same example in the SI, they mention that the effective parameters are chosen such that all moments up to second order of the two systems are matched. It is true that the zeroth- and first-order moment equations depend also on the 2nd moment of the gene in the unbound state, however there is no dependence on the other 2nd moment. I found this detail a bit confusing and perhaps the authors could explain this better in the main text.*

Response: Thank you for pointing out the inconsistency between the main text and the SI. In the revised manuscript, we have provided a more detailed explanation of the Holimap algorithm (see pages 4-7 in the SI). For the 2-HM, we emphasized that the two effective parameters $\tilde{\sigma}_u$ and $\tilde{\sigma}_b$ are chosen so that the nonlinear and linear systems have the same zeroth and first-order moment equations (for the latter, we mean the first-order moment when the gene is in the bound state). For the 4-HM, we emphasized that the four effective parameters $\bar{\sigma}_u$, $\bar{\sigma}_b$, $\bar{\rho}_u$, and $\bar{\rho}_b$ are

chosen so that the two systems have the same zero, first, and second-order moments.

3. *In the same part of the Results, the authors explain how to obtain the effective parameters when the system has not reached steady state. I found that this part was not as clearly explained, so the authors could elaborate a bit more (perhaps in the SI).*

Response: In the revised manuscript, we have provided a more detailed explanation of the Holimap algorithm (see pages 4-7 in the SI).

4. *In several parts of the paper, the authors comment on whether the 2-Holimap or the 4-Holimap is a better choice of approximation. I think it would be useful if the authors made a short summary / table that outlines in what types of scenarios one should choose to use 2-HM over 4-HM or vice versa. On a similar note, I found the hybrid method combining SSA and Holimap interesting and it would be useful to summarise when the hybrid approach should be preferred over a standard Holimap approach. Such a summary would help to better highlight the versatility of Holimap and its hybrid variants in analysing gene networks of different complexities.*

Response: In the revised manuscript, we have summarized which type of Holimap methods should be used in different scenarios (see pages 10-11). In summary, we have devised three types of Holimaps — the 2-HM, 3-HM, and 4-HM — all of them decoupling gene-gene interactions in a nonlinear regulatory network and transforming it into a linear one with multiple effective parameters. The 2-HM and 4-HM apply to transcriptional networks, while the 3-HM is only applicable to post-translational and post-transcriptional networks. The 4-HM is more accurate than the 2-HM, although the improvement in accuracy is marginal. Depending on parameters, the 4-HM may lead to numerical instability at short times. Hence the 4-HM is preferred if our aim is to compute the steady-state distribution, and the 2-HM is a preferable choice if our aim is to compute the time-dependent distribution. The two types of Holimaps require the solution of the same number of moment equations and hence give rise to similar CPU times. Since the number of equations to be solved increases exponentially with the network size, the standard Holimap is only recommended when the scale of the network is not too large. For medium and large-scale networks, the hybrid SSA+Holimap approach is more advantageous since it significantly reduces the CPU time while maintaining high accuracy.

5. *It is suggested that the authors provide documentation in their Github repository, so that the Holimap method is reproducible.*

Response: In the revised manuscript, we have now provided the MATLAB codes for all networks studied in this paper including the random eight-node network, two types of post-translational networks, and two types of post-transcriptional networks. Moreover we have also added a README file in the GitHub repository.

Remarks on code availability

1. *There is no README provided, but it has been suggested to the authors to provide documentation so their method is reproducible.*

Response: The GitHub repository has now been updated to include the README file.

Response to Referee 5

Thank you for your valuable comments and suggestions which have helped us greatly in improving the quality of this paper. Below we provide a point-by-point response to each comment. All modifications made in the revised manuscript are shown in blue.

Response: Thank you for your valuable comments and suggestions.

Reviewer #1 (Remarks to the Author):

Reviewer #2 (Remarks to the Author):

The authors have done substantial work to address my previous comments. The newly added examples on post-translational and post-transcriptional networks improved the study significantly, and will further increase the readership. Other suggestions on textual additions and changes have been taken as well. This revision is adequate in my opinion.

I congratulate the authors on this excellent work!

Reviewer #2 (Remarks on code availability):

The deposited code is helpful for reproducing key results of this work.

Reviewer #3 (Remarks to the Author):

Authors have revised the manuscript a lot in the Results and Discussion as well as Figure 6. They addressed all the points we mentioned in the previous report. I agree with the publication of this manuscript.

Reviewer #4 (Remarks to the Author):

The changes made in the manuscript have addressed my concerns.

Reviewer #4 (Remarks on code availability):

The github page contains scripts that can be used to reproduce the results of the paper.

Reviewer #5 (Remarks to the Author):

I co-reviewed this manuscript with one of the reviewers who provided the listed reports. This is part of the Nature Communications initiative to facilitate training in peer review and to provide appropriate recognition for Early Career Researchers who co-review manuscripts